# YoooP: You Only Optimize One Prototype per Class for Non-Exemplar Incremental Learning

## Abstract

Incremental learning (IL) usually addresses catastrophic forgetting of old tasks when learning new tasks by replaying old tasks' data stored in a memory, which can be limited by its size and the risk of privacy leakage. Recent non-exemplar IL methods only store class centroids as prototypes and perturb them with Gaussian noise to create synthetic data for replay. However, the class prototypes learned in different tasks might be close to each other, leading to the intersection of their synthetic data and forgetting. Moreover, the Gaussian perturbation does not preserve the real data distribution and thus can be detrimental. In this paper, we propose YoooP, a novel exemplar-free IL approach that can greatly outperform previous methods by only storing and replaying one prototype per class even without synthetic data replay. Instead of storing class centroids, YoooP optimizes each class prototype by (1) moving it to the high-density region within every class using an attentional mean-shift algorithm; and (2) minimizing its similarity to other classes' samples and meanwhile maximizing its similarity to samples from its class, resulting in compact classes distant from each other in the representation space. Moreover, we extend YoooP to YoooP+ by synthesizing replay data preserving the angular distribution between each class prototype and the class's real data in history, which cannot be obtained by Gaussian perturbation. YoooP+ effectively stabilizes and further improves YoooP without storing any real data. Extensive experiments demonstrate the superiority of YoooP/YoooP+ over non-exemplar baselines in terms of accuracy and anti-forgetting.

## 1 Introduction

Catastrophic forgetting McCloskey & Cohen (1989) refers to deep neural networks forgetting the acquired knowledge from the previous tasks disastrously while learning the current task. This is in sharp contrast to humans who are able to incrementally learn new knowledge from the ever-changing world. To bridge the gap between artificial intelligence and human intelligence, incremental learning (IL) Wu et al. (2019); Gepperth & Hammer (2016); Douillard et al. (2022); Xie et al. (2022) has emerged as a new paradigm to enable AI systems to continuously learn from new data over time.

In the past few years, a variety of methods Roady et al. (2020); Cong et al. (2020); Wang et al. (2021); Xue et al. (2022) have been proposed to alleviate catastrophic forgetting in IL. In this work, we are interested in a very challenging scenario, called class-incremental learning (CIL) Li & Hoiem (2017); Lopez-Paz & Ranzato (2017); Rebuffi et al. (2017). CIL aims to identify all the previously learned classes with no task identifier available at the inference time. Unfortunately, CIL often suffers from catastrophic forgetting because of the overlapping representations between the previous tasks and the current one in the feature space Zhu et al. (2021b). To deal with this issue, many prior studies adopt *exemplar-based approaches* to preserve some old class samples in a memory buffer. These methods, however, suffer from memory limitations and privacy issues. Thus, some works propose *non-exemplar-based methods* Li & Hoiem (2017); Yu et al. (2020); Lopez-Paz & Ranzato (2017); Mallya & Lazebnik (2018) that incrementally learn new tasks without storing raw samples in a memory. Most of these methods adopt regularization and generative models Li & Hoiem (2017) to mitigate catastrophic forgetting, but they do not perform well in exemplar-free CIL.

Recently, a few studies developed prototype-based methods Zhu et al. (2021b;a); Petit et al. (2023), such as PASS and FeTrIL, that store one prototype (*class mean*) for each old class and then use

augmented prototypes to train a model. Surprisingly, we find that the PASS Zhu et al. (2021b), which augments the stored prototypes via Gaussian noise, will degrade the prediction accuracy compared to that without prototype augmentation, as illustrated in Fig. 1. This is because the class mean prototype may not accurately represent the centroids of different classes in a high-dimensional space. Plus, inappropriate prototype augmentation will cause more confusion between old classes learned in different stages. It thus motivates us to optimize the learning of prototypes for each class in CIL.

In this work, we develop YoooP, a new non-exemplar CIL method that *only needs to store and replay* one class-representative prototype without using synthetic data. The key challenge lies in how to learn a more representative prototype for each class so that these prototypes can be distant from each other. To address this challenge, we propose a new attentional mean-shift based method to optimize the prototypes by aggregating the representations of samples in each class in a high-density region. Different from PASS, we *only replay one stored prototype* for each old class *without using augmented prototypes* during model training. To further improve the accuracy of YoooP, we extend it to YoooP+ which generates synthetic data from stored prototypes. Accordingly, we develop a new prototype augmentation approach that combines a high-dimensional space rotation matrix and the stored angular distribution between each class's prototype and the real data to create synthetic data of old classes. The evaluation

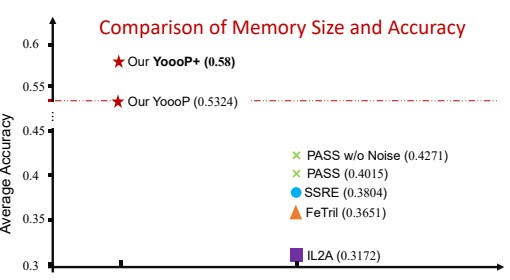

Figure 1: Comparison of memory size (please see Appendix D) and accuracy for different methods on TinyImageNet under zero-base 10 phase setting. The term "memory size" denotes the memory footprint of approaches when computing prototypes for each task during training. Unlike other prototype-based methods that necessitate saving the embeddings of the entire class to generate the class mean prototype, YoooP generates optimized prototypes with a smaller memory footprint by mini-batch attentional mean-shift based method. Moreover, it only stores and replays one prototype for each class can achieve higher accuracy than the baselines.

results on multiple benchmarks demonstrate that both YoooP and YoooP+ can significantly outperform the baselines in terms of accuracy and average forgetting. Moreover, replaying synthetic data via the proposed prototype augmentation can further improve the performance of YoooP.

**Our contributions** are four-fold: 1) we propose a novel non-exemplar CIL algorithm, called YoooP, that can achieve remarkable performance by only replaying one stored prototype for each class without using synthetic data, 2) to our best knowledge, we are the first to explore prototype optimization in CIL, 3) we extend YoooP to YoooP+, which develops a new prototype augmentation technique that creates high-quality data from stored prototypes, 4) the evaluation results demonstrate the superiority of our methods over the non-exemplar baselines in terms of accuracy and average forgetting.

## 2 RELATED WORK

**Regularization-based method.** This method aims to alleviate catastrophic forgetting by introducing additional regularization terms to correct the gradients and protect the old knowledge learned by the model Masana et al. (2022); Li & Hoiem (2017); Rannen et al. (2017); Kirkpatrick et al. (2017); Lee et al. (2017); Liu et al. (2018). Existing works mainly adopt weight regularization to reduce the impact of learning new knowledge on the weights that are important for old tasks. However, it is very hard to design reasonable and reliable metrics to measure the importance of model parameters.

**Parameters isolation-based method.** This line of work can be divided into dynamic network expansion and static network expansion. Dynamic network expansion methods adopt individual parameters for each task, so they need a large memory to store the extended network for each previous task during training Yoon et al. (2017); Ostapenko et al. (2019); Xue et al. (2022); Yan et al. (2021). Conversely, static network expansion approaches Serra et al. (2018); Mallya & Lazebnik (2018); Mallya et al. (2018); Zhu et al. (2022) dynamically expand the network if its capacity is not large enough for new tasks, and then adapt the expanded parameters into the original network. Those methods can achieve remarkable performance, but they are not applicable to a large number of tasks.

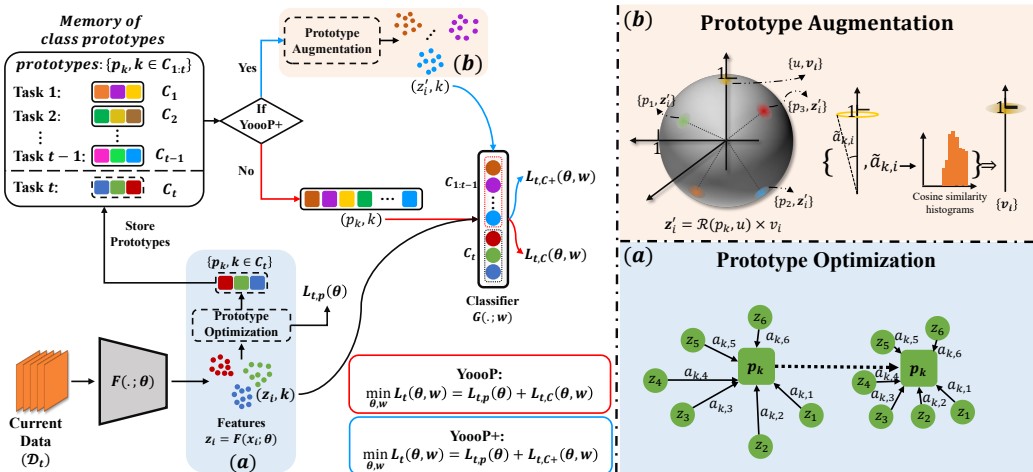

Figure 2: Framework of the proposed YoooP and YoooP+. YoooP only needs to replay one stored prototype for each class while YoooP+ is trained on synthetic data generated from stored prototypes. (a) Prototype optimization learns a compact prototype for each class by a new attentional mean-shift based method (b) Prototype augmentation aims to generate synthetic data of old classes from stored prototypes and angular distribution using a $m$-dimensional space rotation matrix.

**Data replay-based method.** This solution Wu et al. (2018); Cong et al. (2020); Rostami et al. (2019) mainly employs deep generative models to generate synthetic samples of old classes in order to mitigate privacy leakage. Most existing works Shin et al. (2017); Ostapenko et al. (2019); Lesort et al. (2019); Rios & Itti (2018) focus on Variational Autoencoder (VAE) and Generative Adversarial Network (GAN). However, these methods suffer from the instability of generative models and inefficient training for complex datasets.

**Prototype-based method.** Instead of generating pseudo samples, some recent works Zhu et al. (2021b;a); Petit et al. (2023) attempt to store class-representative prototypes and then generate synthetic data via augmentation to enhance the classifier performance and mitigate catastrophic forgetting. The typical works include PASS Zhu et al. (2021b) and its variants, IL2A Zhu et al. (2021a) and FeTrIL Petit et al. (2023). However, the class mean prototype used in these methods may not accurately represent a compact and informative representation for each class. As a result, some augmented prototypes of old tasks may overlap with other similar classes. In contrast, this paper aims to learn a more compact feature space by optimizing the prototype for each class.

## 3  PROPOSED METHOD

In this section, we first elaborate on the proposed YoooP that optimizes the prototype for each class based on an attentional mean-shift method. In order to further improve the prediction accuracy, we extend YoooP to develop a YoooP+ with synthetic data generated from stored prototypes.

**Problem Description.** By learning from a sequence tasks each associated with a subset of classes $C_t$ and a training set of $n_t$ examples drawn from these classes, i.e., $\mathcal{D}_t \triangleq \{x_i, y_i\}_{i=1}^{n_t}$ with $y_i \in C_t$, class-incremental learning (CIL) aims to train a model $f(x; [\theta, w]) \triangleq G(F(x; \theta); w)$ that predicts probabilities of all previous classes $C_{1:t} \triangleq \bigcup_{i=1}^{t} C_i$ for any given example $x$. The model is composed of a feature extractor $F(\cdot; \theta)$ producing compact representations and a classifier $G(\cdot; w)$. Given $x$, the probabilities over all classes $C_{1:t}$ are predicted as softmax$(G(F(x; \theta); w))$.

### 3.1  YoooP

We introduce YoooP, which performs two main steps in each iteration: (i) prototype optimization for learning a compact prototype per class; and (ii) new task learning with prototype replay.

For (i), we propose a novel mini-batch attentional mean-shift based method to optimize the prototype for each class. By prototype optimization, we are able to concentrate most samples to their class prototypes and keep each class a compact cluster in the feature space. This strategy significantly mitigates inter-class interference, which is a primary reason for forgetting.

For (ii), when learning a new task, we augment its training set with previous tasks' class prototypes from the memory. In YoooP, replaying only the memorized prototypes of old classes suffice to retain their classes' features and mitigate catastrophic forgetting.

### 3.1.1 PROTOTYPE OPTIMIZATION

We propose to optimize the prototype of each class based on a mini-batch attentional mean-shift method. Specifically, when learning task-$t$ defined on a set of classes $C_t$, for each class $k \in C_t$, we construct a graph of the class's sample representations $z_i = F(x_i; \theta)$ that connects class-$k$'s prototype $p_k$. Then we try to move $p_k$ towards a high-density region in the feature space. We achieve this by mean-shift of the prototype: we move $p_k$ towards a weighted average over all samples belonging to class-$k$ (their normalized representations in specific) and normalize the new $p_k$, i.e.,

$$p_k \leftarrow (1-\lambda)p_k + \lambda \sum_{i \in [n_t]:y_i=k} a_{k,i} \cdot \frac{z_i}{\|z_i\|_2}, \quad p_k \leftarrow \frac{p_k}{\|p_k\|_2}, \tag{1}$$

where $\lambda$ controls the step size of the mean-shift and $n_t$ is the size of the training set for task-$t$. Unlike the original mean-shift algorithm, the weights $a_{k,i}$ are determined by learnable dot-product attention between each sample $z_i$ and the prototype $p_k$ in the feature space, i.e.,

$$a_k \triangleq \text{softmax}(\bar{a}_k), \quad \bar{a}_k \triangleq [\bar{a}_{k,1}, \cdots, \bar{a}_{k,n_t}], \quad \bar{a}_{k,i} = c(z_i, p_k) \triangleq \frac{\langle z_i, p_k \rangle}{\|z_i\|_2 \cdot \|p_k\|_2}. \tag{2}$$

In practice, when the number of samples $n_t$ is large, we can apply a mini-batch version of Eq. 1 for multiple steps, where $i \in [n_t]$ is replaced by $i \in B$ ($B$ is a mini-batch of samples). We then store the prototype of each class in the memory, which will be used to train the model together with learned tasks' prototypes and a new task's data.

For prototype optimization, we train the representation model $F(\cdot; \theta)$ to produce $z_i = F(x_i; \theta)$ for each sample $x_i$ to be close to its class prototype and distant from other classes' prototypes. We achieve this by minimizing the contrastive loss $\ell_P(\cdot, \cdot, \cdot)$ for task-$t$, i.e.,

$$L_{t,P}(\theta) \triangleq \frac{1}{|C_t|} \frac{1}{n_t} \sum_{k \in C_t} \sum_{i \in [n_t]:y_i=k} \ell_P\left(p_k, z_i, \{p_j\}_{j \in C_{1:t}, j \neq k}\right), \tag{3}$$

This loss aims to minimize the distance between positive pairs $(z_i, p_k)$ while maximizing the distance between negative pairs $(z_i, p_j)$. Hence, samples belonging to each class are enforced to concentrate on their prototype and form a compact cluster distant from other classes in the feature space, which effectively reduces the harmful interference between classes that leads to future catastrophic forgetting.

### 3.1.2 NEW TASK LEARNING WITH PROTOTYPE REPLAY

Meanwhile, in order to mitigate the catastrophic forgetting of previous tasks' classes $C_{1:t-1}$, YoooP replays the stored class prototypes while training with the current task's data, which means we augment the training set for task-$t$ with prototypes from previous classes $C_{1:t-1}$. The corresponding training objective for classification on $C_{1:t}$ is to minimize the negative log-likelihood on all the task-$t$'s data and prototypes for all the previous $t-1$ tasks, i.e.,

$$L_{t,C}(\theta, w) \triangleq \frac{1}{|C_t|} \frac{1}{n_t} \sum_{k \in C_t} \sum_{i \in [n_t]:y_i=k} \ell([c(z_i, w_j)]_{j \in C_{1:t}}, k) + \frac{1}{|C_{1:t-1}|} \sum_{k \in C_{1:t-1}} \ell([c(p_k, w_j)]_{j \in C_{1:t}}, k)$$

$$(4)$$

where $\ell(\cdot, \cdot)$ is a Arcface loss Deng et al. (2019) detailed in the experimental settings, and $i \in [n_t] : y_i = k$ will be replaced by $i \in B$ when training on mini-batch. In summary, $L_{t,P}(\theta)$ mainly focuses on moving the current task's samples to their associated class prototype in the feature space so the

prototypes retain most information of the task. On the other hand, $L_{t,C}(\theta, w)$ trains the representation model's parameter $\theta$ and the classifier layer(s) $w$ in an end-to-end manner on an augmented dataset composed of both the current task's data and the prototypes so the model can learn new tasks without suffering from forgetting previous tasks.

Since CIL requires the model to learn the tasks sequentially, the extractor $F(\cdot; \theta)$ would be updated incrementally. To minimize the drift of the stored prototypes, following previous work Zhu et al. (2021b); Hou et al. (2019), we employ knowledge distillation (KD) Hou et al. (2019) when training $F(\cdot; \theta)$ on the current task data $x \sim \mathcal{D}_t$ by minimizing the difference between $F(x; \theta)$ and the representations $F(x; \theta_{t-1})$ produced by previous task model $\theta_{t-1}$, i.e.,

$$L_{t,KD}(\theta) \triangleq \frac{1}{n_t} \sum_{i \in [n_t]} \|F(x_i; \theta) - F(x_i; \theta_{t-1})\|_2^2. \tag{5}$$

Hence, the training objective $L_t(\theta, w)$ of YoooP at task-$t$ combines the prototype-learning loss for task-$t$ in Eq. 3, the prototype-replay augmented loss in Eq. 4, and the KD loss in Eq. 5 with loss weight $\gamma$, i.e.,

$$\text{YoooP}: \quad \min_{\theta, w} L_t(\theta, w) = L_{t,P}(\theta) + L_{t,C}(\theta, w) + \gamma * L_{t,KD}(\theta). \tag{6}$$

## 3.2 YoooP+

Although prototype-only replay in YoooP is highly effective in mitigating catastrophic forgetting, it might be insufficient to cover all useful information of the whole distribution for each class without replay on different instances. Hence, we propose an extension YoooP+ with the replay of synthetic data augmented from the stored prototypes in the memory.

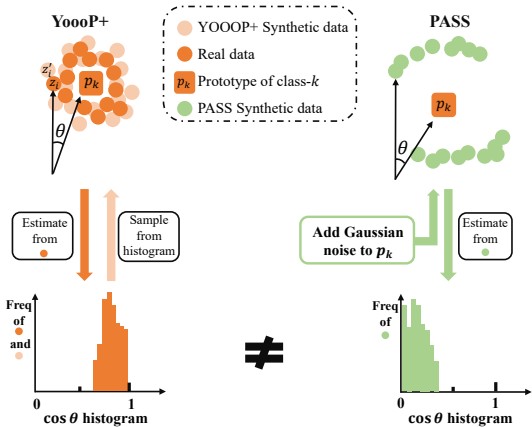

Figure 3: Synthetic data distributions of YoooP+ and PASS with prototype augmentation: **YoooP+ preserves the original angular distribution**.

### 3.2.1 NOVEL PROTOTYPE AUGMENTATION

We propose a new prototype augmentation that draws each class's synthetic data from the angular distribution between each class prototype and the class's real data. To simplify the augmentation, we rotate each prototype to a unit vector before drawing synthetic data and rotate the synthetic data back. As shown in Fig. 3, we sample the cosine similarity from the stored real angular distribution, $P(\bar{a}_{k,i})$, between each class's prototype and the embeddings of real data. This distribution is represented by a histogram with $N_b$ bins. Then we use sampled cosine similarities to generate synthetic data for each class. Consequently, the angular distribution between each class prototype and synthetic data is designed to faithfully preserve the original angular distribution, $P(\bar{a}_{k,i})$. In contrast, approaches like PASS add high-dimensional noise into saved prototypes, causing a significant divergence in the angular distributions between the class prototypes and synthetic data from the actual angular distribution, $P(\bar{a}_{k,i})$.

Specifically, by using the stored $P(\bar{a}_{k,i})$, we are able to synthesize a data point $z_i'$ that has a similar angular distance to the prototype $p_k$ as $z_i$ for replay. This leads to YoooP+ whose replay of each previous class is conducted on multiple synthetic data points instead of a single prototype.

In particular, **we firstly derive a rotation matrix** $\mathcal{R}(p_k, \boldsymbol{u})$ that can recover $p_k$ from a unit vector $\boldsymbol{u} = [1, 0, \cdots, 0]$ on an unit $m$-sphere, i.e., $p_k = \mathcal{R}(p_k, \boldsymbol{u}) \times \boldsymbol{u}$. To synthesize a sample $z_i'$ of class-$k$ as a proxy to $z_i$ (a previously learned sample of class-$k$), **we then randomly draw** $\boldsymbol{v_i}$ in the vicinity of $\boldsymbol{u}$, i.e.,

$$\boldsymbol{v_i} = [\tilde{a}_{k,i}, \epsilon_2, \cdots, \epsilon_m], \quad \tilde{a}_{k,i} \sim P(\bar{a}_{k,i}) \tag{7}$$

To ensure $\|\boldsymbol{v_i}\|_2 = 1$, we draw $\epsilon_i \sim \mathcal{N}(0, 1)$ for $i \in \{2, \cdots, m\}$ at first and then rescale them by $\epsilon_i \leftarrow \sqrt{1 - (\tilde{a}_{k,i} + \epsilon_1)^2} / \sum_{i=2}^m \epsilon_i^2 \cdot \epsilon_i$. Thereby, we have $\boldsymbol{u}^T \boldsymbol{v_i} = \tilde{a}_{k,i}$, whose distribution approximates the distribution of cosine similarity $\bar{a}_{k,i}$ between real sample $z_i$ and its associated class prototype $p_k$.

**Next, we create $z_i'$ from $v_i$.** As $p_k = \mathcal{R}(p_k, \boldsymbol{u}) \times \boldsymbol{u}$, we can apply the same rotation matrix $\mathcal{R}(p_k, \boldsymbol{u})$ to $\boldsymbol{v_i}$ to achieve $z_i'$, i.e.,

$$z_i' = \mathcal{R}(p_k, \boldsymbol{u}) \times \boldsymbol{v_i}. \tag{8}$$

By applying the same rotation, the similarity between $\boldsymbol{u}$ and $\boldsymbol{v_i}$ is preserved between $p_k$ and $z_i'$. By sampling the synthetic data point $z_i'$ for each previously removed sample $z_i$ using the above synthesis, we are able to create a dataset for all seen classes in $C_{1:t}$ that can be used in the replay.

### 3.2.2 NEW TASK LEARNING WITH PROTOTYPE AUGMENTATION

When learning a new task-$t$, YoooP+ also replays the synthetic dataset $\mathcal{D}_t'$ augmented from all previous tasks' prototypes $p_k$, i.e.,

$$\mathcal{D}_t' \triangleq \{(z_i', k) : k \in C_{1:t-1}, z_i' = \mathcal{R}(p_k, \boldsymbol{u}) \times \boldsymbol{v_i}, \boldsymbol{v_i} = [\tilde{a}_{k,i}, \epsilon_2, \cdots, \epsilon_m]\}. \tag{9}$$

The training objective for task-$t$ with the replay of previous tasks' data synthesized from the stored prototypes is

$$L_{t,C+}(\theta, w) \triangleq \frac{1}{|C_t|} \frac{1}{n_t} \sum_{k \in C_t} \sum_{i \in [n_t]: y_i = k} \ell(c(z_i, w), k) + \frac{1}{|\mathcal{D}_t'|} \sum_{(z,k) \in \mathcal{D}_t'} \ell(c(z, w), k). \tag{10}$$

In sum, the training objective $L_t(\theta, w)$ of YoooP+ at task-$t$ combines the prototype-learning loss for task-$t$ in Eq. 3, the synthetic-data replay augmented loss in Eq. 10, and the KD loss in Eq. 5 with loss weight $\gamma$, i.e.,

$$\text{YoooP+}: \quad \min_{\theta, w} L_t(\theta, w) = L_{t,P}(\theta) + L_{t,C+}(\theta, w) + \gamma * L_{t,KD}(\theta). \tag{11}$$

### 3.3 PRACTICAL IMPROVEMENT TO YOOOP/YOOOP+

Finally, we adopt the following techniques to further enhance the model performance.

**Model Interpolation.** We also apply model interpolation to retain the knowledge of the previous model $\theta_{t-1}$ and avoid overfitting to the current task. Specifically, after learning task-$t$, we update the current $\theta_t$ by the following interpolation between $\theta_{t-1}$ and $\theta_t$, i.e.,

$$\theta_t \leftarrow (1 - \beta)\theta_{t-1} + \beta\theta_t, \tag{12}$$

where $\beta \in [0, 1]$ and we set $\beta = 0.6$ in experiments. Since $\theta_t$ is mainly trained on task-$t$, such simple interpolation between $\theta_{t-1}$ and $\theta_t$ leads to a more balanced performance on all tasks.

**"Partial Freezing" of Classifier.** Each row $w_k$ in the classifier parameter $w$ corresponds to a class $k \in C_{1:t}$. Since the current task-$t$ mainly focuses on classes $k \in C_t$, we apply a much smaller learning rate $\eta' \ll \eta$ ($\eta$ is the learning rate for other parameters) to $w_k$ associated with previous classes to avoid significant drift of their classifier parameters, i.e.,

$$w_k \leftarrow w_k - \eta' \nabla_{w_k} L_t(\theta, w), \quad \forall k \in C_{1:t-1} \tag{13}$$

We provide the complete procedure of YoooP and YoooP+ in Algorithm 1 in Appendix A.

## 4 EXPERIMENT

In this section, we first evaluate the performance of the proposed YoooP and YoooP+ on CIFAR-100 Krizhevsky et al. (2009) and TinyImageNet Yao & Miller (2015). Then we evaluate the quality of synthetic data augmented from memorized prototypes. Finally, we do ablation studies to explore the impact of main components and certain hyperparameters on model performance.

**Experimental settings.** We present some important experimental settings as follows. In the experiments, we train the ResNet-18 He et al. (2016) using the SGD Ruder (2016) optimizer with an initial learning rate of 0.01. In order to learn more compact feature space, we follow previous works Sun et al. (2020); Jiao et al. (2019); Meng et al. (2021) to employ the Arcface Deng et al. (2019) in our contrastive loss $\ell_P(\cdot, \cdot, \cdot)$ and classification loss $\ell(\cdot, \cdot)$. Note that the only difference between the Arcface loss and the widely used classification loss is the additive angular margin penalty $\delta$. Then the learning rate is multiplied by 0.1 per 20 epochs. Following the prior work Zhu et al. (2021b;a),

the weight for KD loss should be large. Thus, we choose $\gamma = 30$ in the loss functions in Eq. 6 and Eq. 11in YoooP and YoooP+. The other experimental settings and model configurations are detailed in Appendix B.

**Baselines.** We compare the proposed YoooP and YoooP+ with non-exemplar-based methods, including LwF Li & Hoiem (2017), PASS Zhu et al. (2021b), SSRE Zhu et al. (2022), IL2A Zhu et al. (2021a), and FeTrIL Petit et al. (2023). We measure the performance of different methods with two commonly used metrics in IL: average incremental accuracy Rebuffi et al. (2017) and average forgetting Chaudhry et al. (2018).

## 4.1 EVALUATION RESULTS

First, we assess the performance of the proposed two methods against the baseline approaches on both CIFAR-100 and TinyImageNet datasets.This evaluation covers various phases under the **zero-base and half-base settings**: 5 phases and 10 phases. As shown in Fig. 4, we can observe that both YoooP and YoooP+ outperform all the non-exemplar methods in terms of accuracy under both zero-base setting and half-base setting, except for one special scenario with 10 phrases where FeTril has slightly higher accuracy than our methods on CIFAR-100 under half-base setting. The reason why both YoooP and YoooP+ outperform the PASS is that our prototype optimization can generate a compact prototype in a high-density region for each class, which reduces inter-class interference to mitigate forgetting.

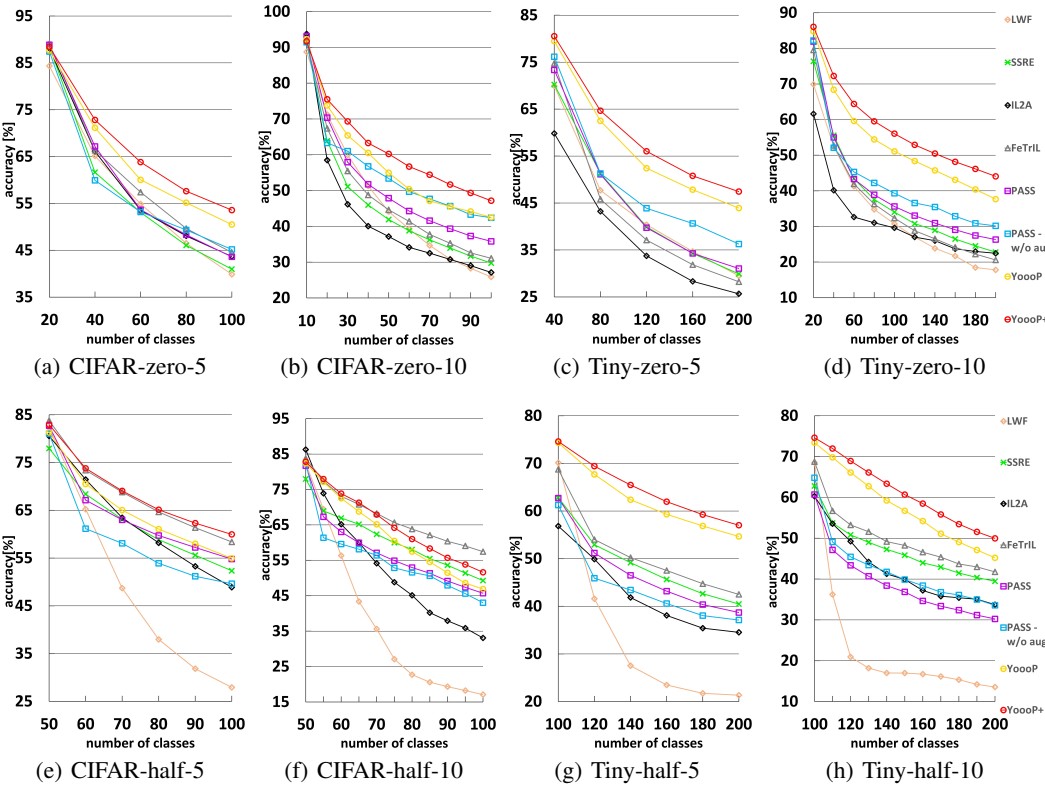

Figure 4: Accuracy comparison of different methods on CIFAR-100 and TinyImageNet under different settings. It can be observed that our YoooP and YoooP+ achieve the best performance in terms of accuracy. "zero-5,10" means "zero-base-5,10 phases" setting and "half-5,10" means "half-base-5,10 phases" setting.

Moreover, we present a comparison of average incremental accuracy and forgetting for different methods, as shown in Table. 1. According to the Table. 1 and upper row of the Fig. 4, both SSRE and FeTrIL have lower forgetting while the prediction accuracy drops rapidly in the initial tasks. A lower forgetting in this case (with lower accuracy) indicates that the model is not improving or learning so such performance is not preferred. In contrast, the proposed YoooP and YoooP+ reach higher average incremental accuracy while achieving slightly higher or comparable forgetting compared to

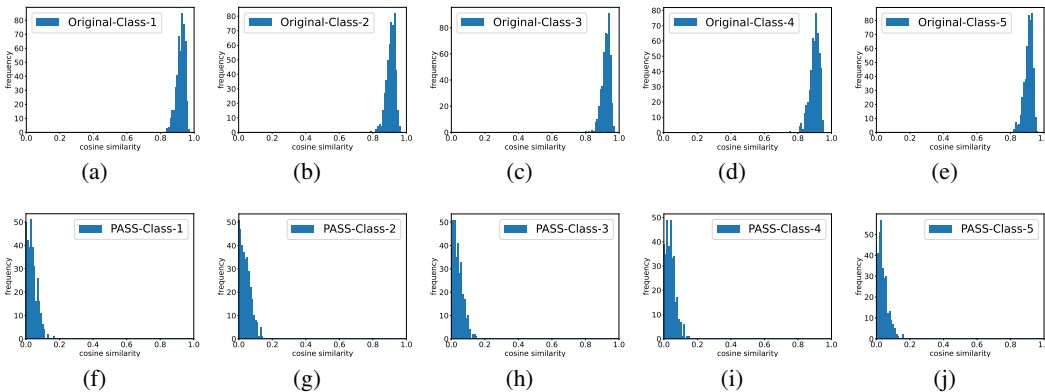

Figure 5: **Top row**: histograms of the original cosine similarity $\bar{a}_{k,i}$ (Eq. 2) between each class's prototype and the real samples (top). The augmented samples of YooP+ are drawn from the original histograms. **Bottom row:** histograms of the cosine similarity between each class's prototype and the augmented samples for PASS. PASS fails to preserve the distribution of the original data.

other non-exemplar baselines. This indicates a better stability-plasticity trade-off, implying that the model learns quickly on new tasks while incurring a minimal cost in forgetting. Therefore, it can be concluded that the proposed methods outperform the non-exemplar baselines.

Table 1: Average incremental accuracy and forgetting of the proposed YoooP and baselines on CIFAR-100 and TinyImageNet under different settings."b0-10" means zero-base with 10 phases, "b0-5" means zero-base with 5 phases. **Bold: the best among non-exemplar methods.**, Red: the second best among non-exemplar methods.

| Average Incremental Accuracy and Forgetting on CIFAR-100 and TinyImageNet | | | | | | | | |
|---|---|---|---|---|---|---|---|---|
| Method | CIFAR-Acc [%]↑ | | CIFAR-Fgt [%]↓ | | Tiny-Acc [%]↑ | | Tiny-Fgt [%]↓ | |
| | b0-5 | b0-10 | b0-5 | b0-10 | b0-5 | b0-10 | b0-5 | b0-10 |
| LwF Li & Hoiem (2017) | 58.15 | 47.43 | 43.80 | 51.80 | 44.44 | 33.8 | 36.77 | 49.15 |
| SSRE Zhu et al. (2022) | 58.05 | 46.58 | **15.44** | **12.13** | 45.13 | 38.04 | 16.31 | 19.94 |
| IL2A Zhu et al. (2021a) | 59.78 | 41.96 | 26.94 | 25.07 | 40.39 | 31.72 | 20.89 | 26.10 |
| FeTrIL Petit et al. (2023) | 61.41 | 48.61 | 18.88 | 16.14 | 43.55 | 36.51 | 15.13 | **15.32** |
| PASS Zhu et al. (2021b) | 60.33 | 51.94 | 23.66 | 18.78 | 45.91 | 40.15 | 18.00 | 16.69 |
| PASS w/o Aug | 59.02 | 55.52 | 28.11 | 29.55 | 48.24 | 42.71 | 24.01 | 26.00 |
| YoooP (Ours) | 64.90 | 57.66 | 17.19 | 16.22 | 57.23 | 53.24 | 14.50 | 20.9 |
| YoooP+ (Ours) | **67.26** | **61.93** | 15.72 | 14.51 | **59.99** | **58.00** | **14.08** | 18.88 |

## 4.2 EVALUATION ON ANGULAR DISTRIBUTION OF SYNTHETIC DATA

Next, we evaluate the angular distribution of synthetic data augmented from stored prototypes in YoooP+ and PASS. In this experiment, we randomly choose five classes in task $t$ from CIFAR-100 when training with 10 phases under zero-base setting. Then we compare two sets of histograms representing the distribution of cosine similarity between the stored prototype and the synthetic or real data for each class, as shown in Fig. 5. The first set (Fig. 5(a)(b)(c)(d)(e)) pertains to the original distribution of cosine similarity between the representations encoded by the extractor $F(\cdot; \theta)$ and the stored prototype for each class. The second set (Fig. 5(f)(g)(h)(i)(j)) concerns the distribution of cosine similarity between the stored prototypes for each class and the associated synthetic data generated by PASS. We can observe that the angular distributions of PASS totally differ from the original distribution. On the other hand, our method samples the cosine similarity in the original angular distribution. As a result, the prototype augmentation in PASS in turn degrades its performance, as illustrated in Fig. 4 and Table. 1. In contrast, YoooP+ augments the stored prototypes with sampled cosine similarity from the original distribution (top row of Fig. 5). This augmentation strategy effectively restores the representations of the original data. Hence, we conclude that YoooP+ can generate more high-quality synthetic data compared to PASS.

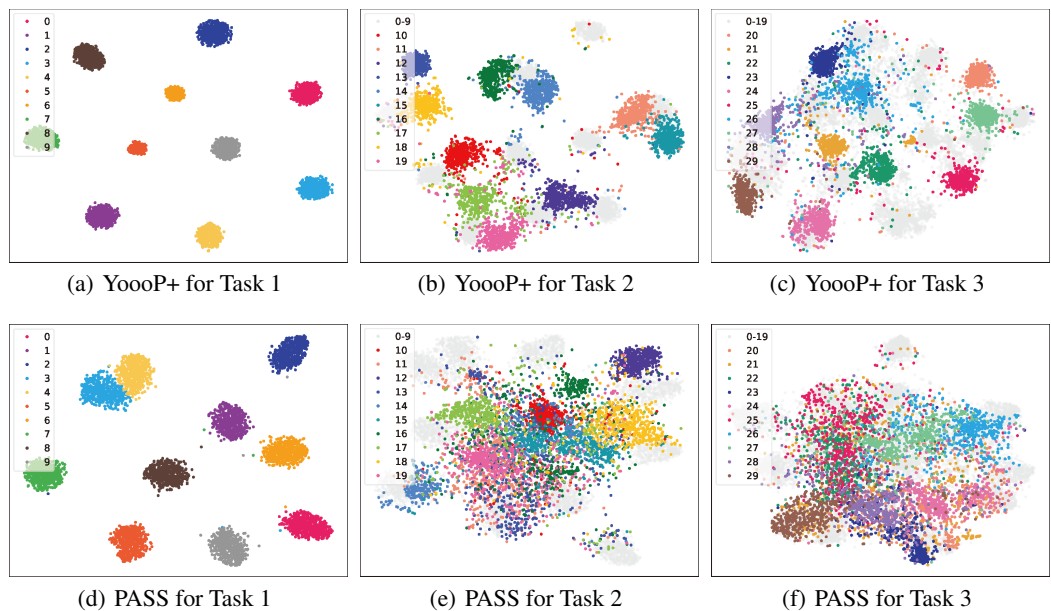

Figure 6: Visualization of the distribution of representations encoded by YoooP+ and PASS on CIFAR-100 base-0 phase 10 setting. The lighter gray points in "Task 2" and "Task 3" represent the distribution of the previous tasks' data.

## 4.3 EVALUATION ON RELIABILITY OF YOOOP+

In addition, we evaluate the reliability of the proposed YoooP+ and PASS in learning the representations from input data samples. Fig. 6 illustrates the distribution of representations encoded by the extractor on the CIFAR-100 under base-0 phase 10 setting for the first three tasks. Both YoooP+ and PASS demonstrate effective maintenance of the decision boundary in the first task. However, in the subsequent tasks 2 and 3, YoooP+ can still encode the input data into a representation within a certain boundary, whereas PASS cannot. In Fig. 6 (b),(c),(e), and (f), the light grey points represent the distribution of data from old tasks. We can see from (b) and (c) that our approach can still separate the old tasks from the current task, while PASS fails to distinguish between the distributions of data from previous tasks and the current one. This is because YoooP+ can form a compact cluster for the samples in each class via prototype optimization and create high-quality synthetic data from the original distribution of cosine similarity to constrain the boundary of old tasks.

## 4.4 ABLATION STUDIES

Finally, we conduct ablation studies to explore the impact of some hyper-parameters and components on the performance of the proposed methods in Appendix C

## 5 CONCLUSION

In this work, we developed two non-exemplar-based methods, YoooP and YoooP+, for class-incremental learning. Specifically, YoooP only needs to store and replay one optimized prototype for each class without generating synthetic data from stored prototypes. As an extension of YoooP, YoooP+ proposed to create synthetic data from the stored prototypes and the stored distribution of cosine similarity with the help of a high-dimensional rotation matrix. The evaluation results on multiple benchmarks demonstrated that both YoooP and YoooP+ can significantly outperform the baselines in terms of accuracy and average forgetting. Importantly, this work offered a new perspective on optimizing class prototypes for exemplar-free CIL. We also show more experiments in Appendix F, and discuss the limitations of this work in Appendix G.

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

## A  YoooP and YoooP+ Algorithms

We summarize the proposed YoooP and YoooP+ algorithms as follows. Regarding parameters $S$ steps and $R$ iterations in Algorithm 1, they are determined by the ratio of Dataset Size to Mini-batch size. More specifically, $S$ is the number of total steps per epoch for each task. For the number of steps $R$ used to update the prototypes in each task, we simply set $R = S$ in our experiments.

---

**Algorithm 1:** YoooP and YoooP+

**input**      :Training data $\mathcal{D}_{1:T}$ with classes $C_{1:T}$, epochs $E$, steps $S$, iterates $R$, number of bins $N_b$
               learning rate $\eta, \eta', \beta$
**initialize** :Memory $\mathcal{M} \leftarrow \emptyset$, Distribution $\mathcal{P} \leftarrow \emptyset, \theta, w$

1  **for** $t = 1 \rightarrow T$ **do**
2     **for** *epoch*=$1 \rightarrow E$ **do**
3        Compute features $z_i = F(x_i; \theta)$ for $(x_i, y_i) \in \mathcal{D}_t$;
4        Compute prototype $p_k$ for every class $k \in C_t$ by iterating Eq. 1 for $R$ iterations;
5        Save prototypes: $\mathcal{M} \leftarrow \mathcal{M} \cup \{(p_k, k) : k \in C_t\}$;
6        Save distribution of cosine similarity using a histogram: $\mathcal{P} \leftarrow \mathcal{P} \cup \{P(\bar{a}_{k,i}) : k \in C_t\}$ ;
7        **for** *step*=$1 \rightarrow S$ **do**
8           Draw a mini-batch of data $(x_i, y_i) \sim \mathcal{D}_t$;
9           **if** *YoooP+* **then**
10             Data Synthesis: create a mini-batch of data $(z_i', k)$ of previous classes from prototypes
               $\{p_k : k \in C_{1:t-1}\}$ and the distribution of cosine similarity $\{P(\bar{a}_{k,i}) : k \in C_t\}$ in
               Eqs. 7,8;
11             Compute loss $L_t(\theta, w)$ in Eq. 11 on the two mini-batches;
12          **else**
13             Draw a mini-batch of prototypes $(p_k, k)$ from $k \in C_{1:t-1}$;
14             Compute loss $L_t(\theta, w)$ in Eq. 6 on the two mini-batches;
15          Update feature extractor: $\theta \leftarrow \theta - \eta \nabla_\theta L_t(\theta, w)$;
16          Update classifier for $k \in C_t$: $w_k \leftarrow \theta - \eta \nabla_{w_k} L_t(\theta, w)$;
17          Update classifier for $k \in C_{1:t-1}$: $w_k \leftarrow \theta - \eta' \nabla_{w_k} L_t(\theta, w)$;
18    Model interpolation: $\theta \leftarrow (1 - \beta)\theta' + \beta\theta$;
19    Save current task model as $\theta' \leftarrow \theta$;
   **output**     :Feature extractor $F(\cdot; \theta)$ and classifier $G(\cdot; w)$

---

## B  Experimental Settings

We implement the proposed methods in PyTorch Paszke et al. (2017) and run the baselines using PyCIL Zhou et al. (2021), which is a well-known toolbox for CIL. In the experiments, we train the ResNet-18 He et al. (2016) using the SGD Ruder (2016) optimizer with an initial learning rate of 0.01. In order to learn more compact feature space, we follow previous works Sun et al. (2020); Jiao et al. (2019); Meng et al. (2021) to employ the Arcface Deng et al. (2019) in our contrastive loss $\ell_P(\cdot, \cdot, \cdot)$ and classification loss $\ell(\cdot, \cdot)$. Note that the only difference between the Arcface loss and the widely used classification loss is the additive angular margin penalty $\delta$. Then the learning rate is multiplied by 0.1 per 20 epochs. Following the prior work Zhu et al. (2021b;a), the weight for KD loss should be large. Thus, we choose $\gamma = 30$ in the loss functions in Eq. 6 and Eq. 11in YoooP and YoooP+. We train the model with batch size 256 for 60 epochs in each task. We set the number of bins $N_b$ to 100 in the histogram to save the distribution of cosine similarity. Specifically, we divide the range $[0, 1]$ into uniform intervals of 100. The experimental results are averaged over three random seeds. We execute different incremental phases (i.e., 5 and 10 phases) under zero-base setting. Specifically, we evenly split the classes of each dataset into several tasks. Following prior work Zhu et al. (2021b), the classes in each dataset are arranged in a fixed random order. As for the memory size of exemplar-based approaches mentioned above, we use *herd election* Rebuffi et al. (2017) to select exemplars of previous tasks under different settings. Specifically, we store 20 samples of each old classes following the settings in Hou et al. (2019); Rebuffi et al. (2017).

## C  ABLATION STUDIES

### C.1  EFFECT OF IMPORTANT COMPONENTS

We first study the impact of some important components, such as prototype optimization, synthetic data replay, and model interpolation (MI), on prediction accuracy. Fig. 7 illustrates the comparison results on CIFAR-100 under 10 phases setting. Here YoooP (Class Mean) represents YoooP adopts the class mean like PASS but with prototype optimization loss in Eq. 3, YoooP (-P) denotes YoooP without prototype optimization, YoooP (-MI) means YoooP without using MI, and YoooP (-P-MI) indicates YoooP does not include prototype optimization and MI.

We can observe from the figure that prototype optimization plays an important role in our proposed methods. The prediction accuracy will drop a lot without it, as seen in YoooP (-P). Comparing YoooP with YoooP (Class Mean), we can find that the proposed YoooP achieves higher accuracy than YoooP (Class Mean) and also reduces *memory footprint* (please refer to Appendix D). Additionally, YoooP+ with novel prototype augmentation can also improve prediction accuracy compared to YoooP. Besides, MI helps to improve model performance since it can retain the knowledge of the prior model while ensuring the good performance of the current model.

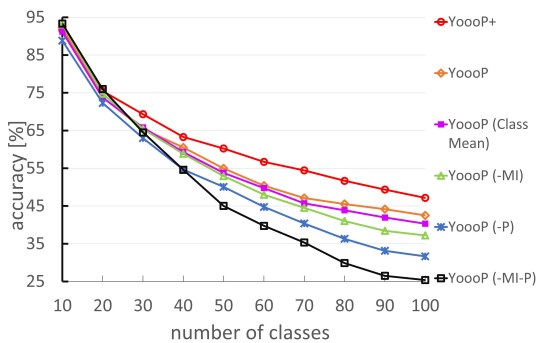

Figure 7: Ablation study of different components in YoooP+. "-MI" means without model interpolation, "-P" means without prototype optimization.

In addition, we conduct more ablation studies to explore some important hyper-parameters on model performance.

### C.2  EFFECT OF $\lambda$ IN MEAN-SHIFT METHOD

Table. 2 shows the impact of hyper-parameter $\lambda$ in mean-shift on prototype learning using CIFAR100. We can observe from it that as $\lambda$ increases from 0.3 to 0.9, the prediction accuracy of YoooP+ almost remains the same. Thus, we can conclude that the proposed method is not sensitive to $\lambda$ that controls the steps in the attentional mean-shift algorithm. We choose $\lambda = 0.6$ in our experiments.

Table 2: Effect of $\lambda$ on the accuracy of different tasks on CIFAR-100 with base-0 10 phases setting.

| Base-0-10 phases Accuracy on CIFAR-100 with different $\lambda$ | | | | | | | | | | |
|---|---|---|---|---|---|---|---|---|---|---|
| $\lambda$ | Task 1 | Taks 2 | Task 3 | Task 4 | Task 5 | Task 6 | Task 7 | Task 8 | Task 9 | Task 10 |
| 0.3 | 0.925 | 0.6855 | 0.6427 | 0.5830 | 0.5840 | 0.5425 | 0.5279 | 0.4959 | 0.4701 | 0.4457 |
| 0.4 | 0.9136 | 0.7433 | 0.6813 | 0.6396 | 0.6037 | 0.5652 | 0.5335 | 0.5003 | 0.4690 | 0.4476 |
| 0.5 | 0.926 | 0.6900 | 0.6450 | 0.5936 | 0.5789 | 0.5376 | 0.5189 | 0.4862 | 0.4628 | 0.4333 |
| 0.6 | 0.9134 | 0.7430 | 0.6810 | 0.6417 | 0.6055 | 0.5667 | 0.5359 | 0.5022 | 0.4706 | 0.4478 |
| 0.7 | 0.9128 | 0.7455 | 0.6851 | 0.6422 | 0.6049 | 0.5663 | 0.5365 | 0.5026 | 0.4691 | 0.4473 |
| 0.8 | 0.9140 | 0.7459 | 0.6825 | 0.6426 | 0.6059 | 0.5675 | 0.5370 | 0.5024 | 0.4706 | 0.4481 |
| 0.9 | 0.9135 | 0.7463 | 0.6823 | 0.6426 | 0.6063 | 0.5682 | 0.5362 | 0.5025 | 0.4689 | 0.4474 |

### C.3  EFFECT OF PARAMETER $\beta$ IN MODEL INTERPOLATION

In addition, we investigate the influence of hyperparameter $\beta$ in model interpolation on the prediction accuracy, as shown in Fig. 8 (a). It can be seen that when $\beta = 0.6$, the proposed method has the best performance. We change the $\beta$ from 0.3 to 0.8, the performance just slightly drops when the $\beta = 0.8$. Thus we conclude the proposed method is not sensitive to $\beta$.

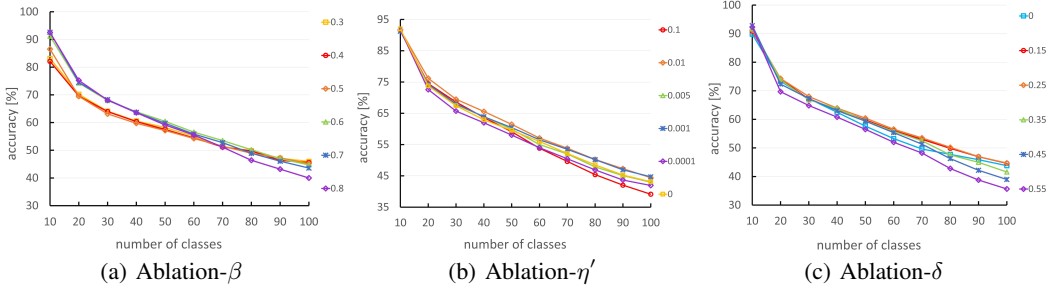

(a) Ablation-$\beta$          (b) Ablation-$\eta'$          (c) Ablation-$\delta$

Figure 8: The ablation study of $\delta$, $\beta$ and $\eta'$ on the prediction results.

### C.4 EFFECT OF SMALL LEARNING RATE $\eta'$

Moreover, we study the effect of learning rate $\eta'$ in the "Partial Freezing" of the classifier on the model performance. As illustrated in Fig. 8 (b), it can be observed that when $\eta'$ is 0 or larger than 0.005, the prediction accuracy will drop slightly. As $\eta'$ is a very small value, such as 0.001, the proposed method has the best performance. Hence, we choose $\eta' = 0.001$ in our experiment.

### C.5 EFFECT OF MARGIN PENALTY $\delta$.

We also investigate the effect of margin penalty $\delta$ in the Arcface in the loss function on the prediction accuracy of our method. It can be observed from Fig. 8 (c) that as $\delta$ increases from 0 to 0.55, the performance almost does not change too much and the model achieves great performance when the $\delta = 0.25$. According to this, we choose $\delta = 0.25$ in our experiments, and we conclude the proposed method is not sensitive to $\delta$.

## D EVALUATION ON MEMORY SIZE

We further analyze the memory usage of the proposed mini-batch attentional mean-shift method presented in Eq. 1. This analysis involves comparing the memory footprint of our method and other approaches when generating prototypes for individual tasks under different settings on TinyImageNet dataset. Note that LwF does not utilize and store the prototype for each class, we thus omit it in our comparison. As shown in Table. 3, our proposed approach can generate the prototypes with much smaller memory footprint than the class mean method used in the baselines. This is because the memory cost of our method relies only on the batch size while the baseline methods need to store the representations of all samples for each class.

Table 3: Memory cost for generating prototype in each task of our proposed method and class mean method in the baselines under different settings on TinyImageNet.

| | Used memory size under different settings on TinyImageNet (MB) ↓ | | | | | |
|---|---|---|---|---|---|---|
| Method | Our method | | class-mean | | | |
| | batch size=128 | batch size=256 | PASS | FeTrIL | IL2A | SSRE |
| base0-phases5 | **0.25** | 0.5 | 39.06 | 39.06 | 39.06 | 39.06 |
| base0-phases10 | **0.25** | 0.5 | 19.53 | 19.53 | 19.53 | 19.53 |
| half-base-phases5 | **0.25** | 0.5 | 19.53 | 19.53 | 19.53 | 19.53 |
| half-base-phases10 | **0.25** | 0.5 | 9.76 | 9.76 | 9.76 | 9.76 |

## E EVALUATION ON THE CONFUSION MATRIX

To effectively demonstrate the results of Class-Incremental Learning (CIL), we conducted a comparison of the classification confusion matrices for PASS, YoooP, and YoooP+ on CIFAR-100 under the zero-base 10-phase setting. The confusion matrix, as shown in Fig. 9, illustrates correct predictions on the diagonal and misclassifications in off-diagonal entries.

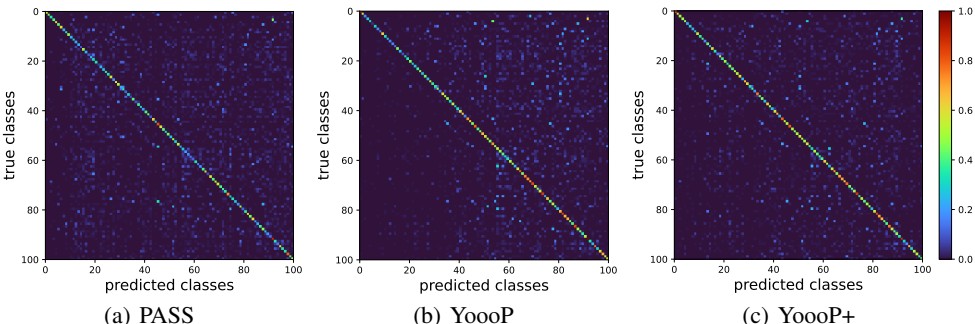

(a) PASS             (b) YoooP             (c) YoooP+

Figure 9: Confusion matrix comparison of different methods under CIFAR-100 base0-10 setting.

Comparing Fig.9(a) and Fig.9(b), we observe that the diagonal entries' temperatures in YoooP are higher than those in PASS, which means the accuracy is higher, particularly for classes 20 to 100. This indicates that YoooP excels in learning the current tasks while minimizing forgetting of the old tasks. Consequently, YoooP exhibits enhanced plasticity and stability compared to PASS.

Further comparing Fig.9(b) and Fig.9(c), we notice that the diagonal entries' temperatures for classes 0 to 40 in YoooP+ are higher than those in YoooP. Notably, YoooP tends to classify more into recent tasks. YoooP+ effectively mitigates this tendency and reduces bias through the proposed prototype augmentation, resulting in superior performance.

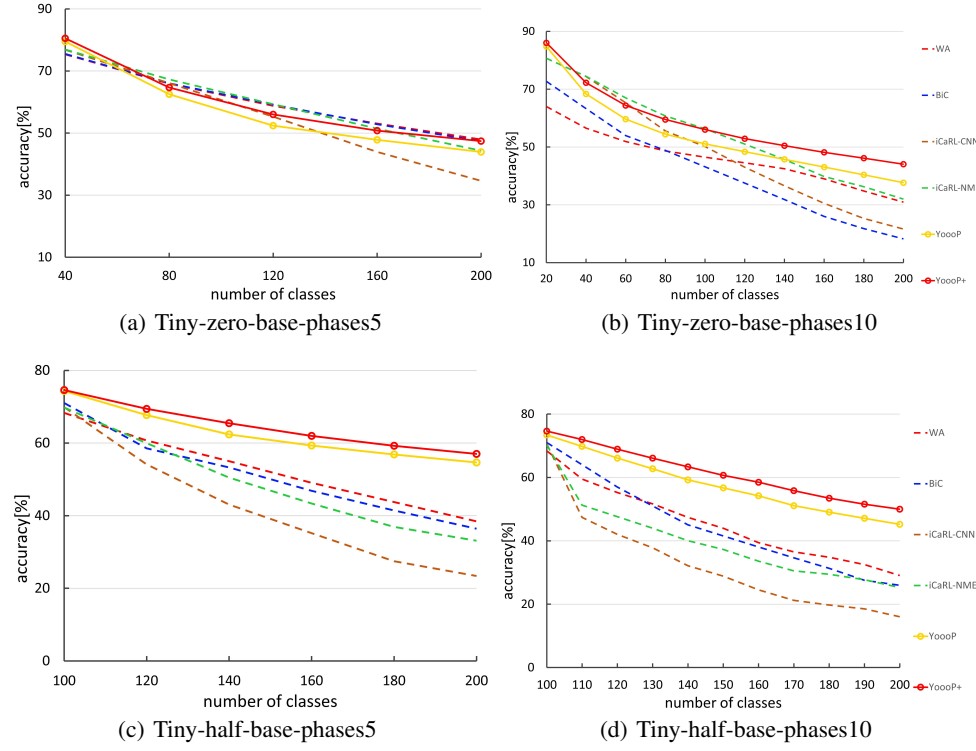

(a) Tiny-zero-base-phases5          (b) Tiny-zero-base-phases10

(c) Tiny-half-base-phases5          (d) Tiny-half-base-phases10

Figure 10: Accuracy comparison of different exemplar-based methods on TinyImageNet under different settings using 3 random seeds.

## F  COMPARE WITH EXEMPLAR-BASED METHODS

To show the great performance of the proposed methods, we also compare the proposed methods with some exemplar-based methods, including iCaRL Rebuffi et al. (2017), BiC Wu et al. (2019), and WA Zhao et al. (2020), under zero-base and half-base settings with 5 and 10 phrases on TinyImageNet. Fig. 10 illustrates the accuracy comparison of different approaches on TinyImageNet using three

random seeds. We can observe that the proposed YoooP and YoooP+ can achieve comparable results under the zero-base settings, and even outperform some exemplar-based methods under the half-base settings.

## G    LIMITATIONS AND FUTURE WORK

This work does not select the synthetic data generated from memorized prototypes for each class, so a few synthetic data will be mixed with those from other classes in the representation space. For future work, we plan to adopt K-nearest neighbor (KNN) to select the appropriate synthetic data for each class in order to further improve the prediction accuracy.

