# OpenReview forum: "YoooP: You Only Optimize One Prototype per Class for Non-Exemplar Incremental Learning"
_ICLR.cc/2024/Conference — Submitted to ICLR 2024_

### Official Review · Reviewer_EwLK · 2023-10-22

**Soundness:** 3 good
**Presentation:** 3 good
**Contribution:** 3 good
**Rating:** 6
**Confidence:** 5

**Summary:**

This paper tackles the non-exemplar class-incremental learning problem, an important problem in the machine learning field. The authors propose the extension of the prototypical network as corresponding loss terms to tackle this problem. The proposed method is evaluated on several benchmark datasets against other competitors.

**Strengths:**

1. This paper tackles the non-exemplar class-incremental learning problem, an important problem in the machine learning field.
2. The proposed method is evaluated on several benchmark datasets against other competitors.
3. The ablation study and visualizations are clear and intuitive.

**Weaknesses:**

1. Overall, I find the title may be unsuitable for the current manuscript. The proposed method utilizes prototypes to construct the training loss, while the embedding and classifiers are always optimizable throughout the learning process. Hence, not only “prototypes” are optimized, but also the backbone and classifiers are optimized for the current task.
2. Some illustrations are unclear. For example, there is no X coordinate in Figure 1, making it hard to figure out the memory usage of different methods. Corresponding explanations are also needed to show why other methods like PASS and FeTrIL require a much larger memory size. These methods also save the class prototype to generate instances, and the memory gap between them should be illustrated.
3. The experimental results should be reorganized. The current results in the main paper only focus on the Base-0 setting, while I also see the results in supplementary that some Base-50 results are also available. It would be better to reorganize these results to contain both settings since Base 50 is also common in today’s CIL.

**Questions:**

1. Please explain the significance of sampling in Section 3.2.1 and its advantage over PASS/FeTrIL. Besides, clarifying the memory budget of different methods in Figure 1 and Section D is also essential. Showing a table with the extra memory budget could be a good solution.
2. It requires further experimental analysis, e.g., the influence of hyper-parameter gamma in Eq. 11.

In summary, this paper tackles an interesting problem with novel techniques. The proposed method shows competitive results in the benchmark comparison. Generally, I am positive about this submission, while addressing the concerns above is also essential for my decision.

---

> ### Author Response · Authors · 2023-11-17
> **Response to Reviewer EwLK**
>
> Thank you for your constructive comments and suggestions! We have addressed all the comments and suggestions from the Reviewer and accordingly updated our manuscript highlighted in BLUE. We hope our responses below address your concerns. Please let us know if you have any additional concerns.
>
>
> Q1: The title may be unsuitable.
> -----------------
> **A1:** Thanks for your suggestion! We will change the title to
> "OPT: Optimizing Prototypes for Non-Exemplar Class-Incremental Learning" if the reviewer thinks it is appropriate.
>
> Q2: Some illustrations are unclear.
> -----------------
> **A2:** We have added the X coordinate for Figure 1 and also explained the "memory footprint" in the caption in our revised manuscript.
>
> Additionally, we have explained the memory gap between our method and existing studies in the revised version. As described in the caption of Figure 1 and detailed in Appendix D, the term "memory size" denotes the memory footprint of our method and other approaches when computing prototypes for each task during training. Unlike other prototype-based methods that necessitate saving the embeddings of the entire class to generate the class mean prototype, YoooP generates optimized prototypes with a smaller memory footprint based on a mini-batch attentional mean-shift algorithm. That is why our approach requires a smaller memory footprint than the existing methods.
>
> Q3: The experimental results should be reorganized.
> -----------------
> **A3:** Per your suggestion, we have moved the experimental results under half-base 5 phases and half-base 10 phases settings to the main paper.
>
> Q4: Explain the significance and advantage of sampling in Section 3.2.1.
> -----------------
> **A4:** Per your suggestion, we have added some explanations about sampling in Section 3.2.1. The main advantage of our method is that we store the distribution of real angular using a histogram while existing works did not. The existing approaches like PASS involve adding high-dimensional noise into prototypes,  causing a significant divergence in the angular distributions between the class prototypes and synthetic data from the actual angular distribution. Moreover, Figure 5 shows the angular distribution comparison of synthetic data augmented from stored prototypes in PASS and the ground-truth angular distribution. We can observe that the angular distributions of PASS differ from the original distribution. On the other hand, our method samples the cosine similarity in the original angular distribution. Hence, our method can generate higher-quality synthetic data than the baselines like PASS and FeTrIL, contributing to a more effective mitigation of forgetting. For more details, please refer to Sections 3.2.1 and 4.2 in our revised version.
>
> Q5: Clarifying the memory budget.
> -----------------
> **A5:** Per your suggestion, we have presented the additional memory requirements of various methods for computing prototypes in Table 1. Here, the term 'extra memory budget' specifically denotes the number of saved embeddings utilized for creating class prototypes. In our methods, this corresponds to the number of embeddings in a mini-batch. In contrast, the other methods necessitate storing the entire set of embeddings for each task to calculate class mean prototypes.
>
> Table 1: Extra memory budget: # of saved embeddings
> | TinyImageNet | Our(BS=128) | Our(BS=256) | PASS | FeTrIL | IL2A | SSRE |
> | -------- | -------- | -------- | -------- | -------- | -------- | -------- |
> | base0-phases5 | 128 | 256 | 20000 | 20000 | 20000 | 20000 |
> | base0-phases10| 128 | 256 | 10000 | 10000 | 10000 | 10000 |
> | half-base-phases5 | 128 | 256 | 10000 | 10000 | 10000 | 10000 |
> | half-base-phases10 | 128 | 256 | 5000 | 5000 | 5000 | 5000 |
> | | | | | | | |
>
>
> Q6: The influence of hyper-parameter $\gamma$ in Eq. 11.
> -----------------
> **A6:** We have explored the impact of hyper-parameter $\gamma$ on CIFAR-100 with zero-base 10 phases setting, as illustrated in the following table. It can be observed that the hyper-parameter $\gamma$ is not very sensitive, and increasing the weight of knowledge distillation within a certain range can be beneficial for YoooP/YoooP+. We can see that weight values between 25 to 40 result in excellent performance.
>
> Table 2: Influence of hyper-parameter $\gamma$ on model performance
> | $\gamma$ | 15 | 20 | 25 | 30 | 35 | 40 |
> | -------- | -------- | -------- | -------- | -------- | -------- | -------- |
> | YoooP+ (Avg) | 60.69\% | 60.44\% | 61.18\% | 61.93\% | 61.02\% | 61.30\% |
> | YoooP+ (last)| 41.21\% | 42.17\% | 44.48\% | 47.17\% | 45.79\% | 46.51\% |
> |   |      |      |        |       |       |     |

---

> > ### Author Response · Authors · 2023-11-21
> > **Reminder to reviewer: Looking Forward to Your Further Comments and Feedback**
> >
> > Dear Reviewer EwLK,
> >
> > We would like to thank you for taking the time to review our paper and for the insightful comments. We have addressed all the comments and suggestions you made. In particular, as suggested, we have **reorganized the experimental results, explained the advantage of sampling, as well as presented the additional memory requirements of various methods for computing prototypes in our response**. As we are approaching the midpoint of the discussion period, please kindly let us know if you have any additional concerns. We truly appreciate this opportunity to improve our work and shall be most grateful for any feedback you could give to us. If you do not have further questions, we are curious if you could consider raising our score. Thank you very much!

---

> > > ### Comment · Reviewer_EwLK · 2023-11-22
> > >
> > > I thank the authors for providing a rebuttal. For now, the details about the "Memory Size" in Figure 1 are made clear, which refers to the "footprint" of prototype calculation. However, it seems the "footprint" is not a problem that requires such a highlight since only one prototype is maintained for each class for all methods, and the "true" memory cost is not so much (but the same for all methods). From this perspective, I suggest not highlighting such a characteristic since it is only about the way of prototype calculation (batch-wise versus class-wise).
> > >
> > > I am satisfied with the responses to my other concerns. I'm somewhat on the fence because other reviewers also raised many concerns about this paper (and part of them make sense to me). Currently, I tend to maintain my initial rating of a "6", but my decision will finally depend on the discussion among reviewers.

---

> > > > ### Author Response · Authors · 2023-11-23
> > > > **Response to Reviewer EwLK**
> > > >
> > > > Thank you for your response and helpful suggestion! We are glad to hear that your concerns have been successfully addressed and you are satisfied with our responses!

---

### Official Review · Reviewer_eHC3 · 2023-10-30

**Soundness:** 2 fair
**Presentation:** 3 good
**Contribution:** 2 fair
**Rating:** 3
**Confidence:** 4

**Summary:**

This paper proposed a prototype-based method that can concentrate most samples to their class prototypes and keep each class a compact cluster in the feature space, which can mitigate inter-class interference. When learning a new task, the saved prototypes are used for replay to deal with forgetting.

**Strengths:**

** The proposed method is reasonable.
** The paper is easy to read.

**Weaknesses:**

** The method is incremental as some prototype-based methods already exist. The paper made some improvements. But the improvements can also be obtained by contrastive learning as the proposed idea is very similar to that of contrastive learning.
** The paper didn’t say where the forgetting occurs. My understanding is that the technique simply learns and saves the prototype for each class. Each task is learned independently. Then there is no forgetting.
** The inference procedure is not discussed. If the algorithm saves only the class prototype for each class, how does it get the feature representation for each test sample for cosine similarity computation in prediction?
** The average accuracy in (Rebuffi et al. 2017) is average incremental accuracy (AIC). Based on AIC, your reported accuracy results are low. Please also report the last accuracy, i.e., after learning the last task and compare with the above methods.
** The experimental datasets and baselines are too few. More baselines with or without using exemplars should be compared as saving some data is not an issue. [a] is a non-exemplar based method.
(1) Kim et al. (2022). A theoretical study on solving continual learning. NeurIPS.
(2) Wang et al. (2022). Beef: Bi-compatible class-incremental learning via energy-based expansion and fusion. ICLR.
(3) Wu et al. (2019) Large scale incremental learning. CVPR.
(4) Buzzega et al. (2020) Dark experience for general continual learning: a strong, simple baseline. NeurIPS.
** Please give the efficiency results. The method seems to be quite slow.

**Questions:**

see the weaknesses

---

> ### Author Response · Authors · 2023-11-17
> **Response to Reviewer eHC3 (Part 1/3)**
>
> Thank you for your constructive comments and suggestions! We have addressed all the comments and suggestions from the Reviewer and accordingly updated our manuscript highlighted in BLUE. We hope our responses below address your concerns. Please let us know if you have any additional concerns.
>
>
>
> Q1: Idea is similar to contrastive learning.
> -----------------
> **A1:** Our approaches bring substantial advancements over existing prototype-based methods. Unlike existing prototype-based approaches that only calculate the **mean prototype** for each class, we propose a novel attentional mean-shift algorithm for optimizing the prototype per class. The proposed method helps to learn a more compact and informative prototype. To our best knowledge, we are the first to study prototype optimization in CIL. Moreover, we develop a novel prototype augmentation technique based on a high-dimensional rotation matrix to generate high-quality synthetic data. In contrast, existing studies only add Gaussian noise into saved prototypes for data generation. The evaluation results demonstrated our methods can significantly outperform the baselines on multiple datasets under different settings. Thus, other reviewers, such as Reviewer #ZsAp, gcwm, and EwLK, recognize the novelty of our proposed methods.
>
> In addition, we would like to highlight the distinction between our method and contrastive learning. Contrastive learning is known as self-supervised learning for pre-training. It aims to learn the representations of input data through the comparison of sample pairs, rather than learning a signal from individual data samples sequentially. This comparison involves both positive pairs (comprising "similar" inputs) and negative pairs (involving "dissimilar" inputs) [1]. In contrast, our methods introduce a novel attentional mean-shift algorithm for optimizing the prototype for each class. We then apply contrastive loss to further enhance the representativeness of each class's prototype. These two components ensure that the learned prototypes are more effectively representative. Note that the optimized prototypes can not be directly obtained through contrastive learning. Rather, the proposed attentional mean-shift algorithm plays a critical role here, though the contrastive loss is helpful.
>
> **References**
>
> [1] Contrastive representation learning: A framework and review. (Ieee Access 2020)
>
>
> Q2: Where the forgetting occurs?
> -----------------
> **A2:** As mentioned in our manuscript, catastrophic forgetting [1] refers to deep neural networks forgetting the acquired knowledge from the previous tasks disastrously while learning the current task. In other words, forgetting will occur when neural networks learn new tasks in sequence. In the context of Class-Incremental Learning (CIL), each task is learned independently and sequentially using **a single shared neural network**. Since all the tasks share the same network, learning a new task can overwrite the network weights learned for the previous tasks [2]. This leads to forgetting, i.e., accuracy drop on previous tasks.
>
> **References**
>
> [1] Catastrophic interference in connectionist networks: The sequential learning problem. (Psychology of learning and motivation 1989, p109-165)
>
> [2] Prototype Augmentation and Self-Supervision for Incremental Learning. (CVPR2021)
>
> Q3: The inference procedure is not discussed.
> -----------------
> **A3:** In Class-Incremental Learning (CIL), the inference for each task is performed similarly to a standard classification task. Specifically, as depicted in Figure 2, we use the backbone, denoted as $F(;\theta)$, to extract embeddings from the test samples. These embeddings are then fed into the classifier $G(;w)$ to estimate the probability distribution over all classes. Conversely, the saved prototypes are merely utilized during the model training phase to avoid forgetting.

---

> ### Author Response · Authors · 2023-11-17
> **Response to Reviewer eHC3 (Part 2/3)**
>
> Q4: Report the last accuracy and compare with iCaRL.
> -----------------
> **A4:** Thanks for your suggestion! Our average accuracy means the average incremental accuracy (AIA) in our submission! Per your suggestion, we already updated the name of this metric in the revised manuscript.
>
>
> Note that our proposed models are **non-exemplar CIL methods** that do not store real samples for replay but only store prototypes, while the iCaRL is an **exemplar-based method** that needs to store and replay real samples. Thus, iCaRL requires a much larger memory budget than our methods. Moreover, prior studies [3][4][5] have pointed out that exemplar-based CIL may suffer from extra risk of privacy leakage and large memory costs. In reality, non-exemplar CIL is much more challenging than exemplar CIL as mentioned by the Reviewer gcwm. Therefore, a direct comparison between our method and iCaRL seems unfair.
>
> Nevertheless, we already compared our methods with iCaRL [1] on the TinyImageNet dataset under different settings in Figure 10 in Appendix F. We report the last accuracy in the following table. It can be observed that our methods still outperform the iCaRL in terms of average incremental accuracy.
>
> Table 1: Evaluation of TinyImageNet.
> |  TinyImageNet  | zero-5 last | zero-10 last | Half-5 last | Half-10 last |
> |  ----  | ----  | ---- | ---- | ---- |
> | * YoooP | 43.93\% | 37.67\% | 54.67\% | 45.22\% |
> | * YoooP+ | **47.44**\% | **44.06\%** | **57.02\%** | **49.97\%** |
> | iCaRL-CNN | 34.65\% | 21.64\% | 23.39\% | 16.04\% |
> | iCaRL-NME | 44.33\% | 31.96\% | 33.10\% | 25.27\% |
> | BiC | 47.41\% | 18.22\% | 36.41\% | 25.98\% |
> | WA | 47.96\% | 30.94\% | 38.45\% | 29.07\% |
> |      |         |      |      |       |
>
> **References**
>
> [1] Incremental Classifier and Representation Learning. (CVPR2017)
>
> [2] Feature Translation for Exemplar-Free Class-Incremental Learning. (WACV2023)
>
> [4] Prototype Augmentation and Self-Supervision for Incremental Learning. (CVPR 2021)
>
> [5] Self-Sustaining Representation Expansion for Non-Exemplar Class-Incremental Learning. (CVPR 2022)
>
> [6] Feature Translation for Exemplar-Free Class-Incremental Learning. (WACV 2023)
>
>
>
> Q5: More datasets and baselines with or without using exemplars should be compared as saving some data is not an issue.
> -----------------
> **A5:** Per your suggestion, we added a new experiment evaluating our proposed methods on another large ImageNet-100 dataset. The comparison results are illustrated in the table below. Our methods (YoooP and YoooP+) still significantly outperform the baselines in terms of average incremental accuracy on the new ImageNet-100 dataset.
>
> Table 2: Evaluation of ImageNet-100.
> |  ImageNet-100  | Half-10 Avg | Half-10 last |
> |  ----  | ----  | ---- |
> | * YoooP | 75.97\% | 68.96\% |
> | * YoooP+ | 77.64\% | 70.54\% |
> | * PASS | 64.3\% | 53.54\% |
> | * FeTril | 72.55\% | 64.06\% |
> | * SSRE | 60.23\% | 51.20\% |
> | * LWF | 47.35\% | 30.53\% |
> | iCaRL-CNN | 48.46\% | 36.29\% |
> | iCaRL-NME | 59.04\% | 48.69\% |
> | BiC | 50.69\% | 31.06\% |
> | WA | 61.04\% | 46.24\% |
> |     |       |
>
> Regarding more baselines, the proposed models are **non-exemplar CIL methods** that do not store real samples for replay but only store prototypes, while the baselines you suggested are **exemplar-based methods** that need to store and replay real samples. Thus, comparing non-exemplar methods with the latest exemplar methods could be deemed unfair. Moreover, prior studies [1][2][3] have pointed out that exemplar-based CIL may suffer from extra risk of privacy leakage and large memory costs. Nonetheless, we still compared our method with a few exemplar-based methods like BiC [4], iCaRL [5], and WA [6] in Figure 11 in Appendix G. We can see that the proposed YoooP and YoooP+ can achieve comparable results under the zero-base settings, and even outperform some exemplar-based methods under the half-base settings.
>
> Concerning the previous study [7], they utilize task ID information during training, which deviates from the established definition of Class-incremental learning (CIL). As outlined in [8] and [9], CIL, true to its name, operates without predefined task boundaries. Consequently, CIL methods typically lack access to any task ID information during both training and inference phases. This implies that it's impractical to train a model to predict the task ID of input samples during testing or inference in CIL scenarios. Unlike prior work [7], we only access the class label rather than task ID information during model training, which aligns with the methodology of prior works [1],[2],[3],[4],[5],[6]. Therefore, we have opted to exclude that particular work [7] from our comparative analysis.
>
> **References (continue)**
>
> [1] Prototype Augmentation and Self-Supervision for Incremental Learning. (CVPR 2021)
>
> [2] Self-Sustaining Representation Expansion for Non-Exemplar Class-Incremental Learning. (CVPR 2022)
>
> [3] Feature Translation for Exemplar-Free Class-Incremental Learning. (WACV 2023)

---

> > ### Author Response · Authors · 2023-11-17
> > **Response to Reviewer eHC3 (Part 3/3)**
> >
> > **References (continue for the above A5)**
> >
> > [4] Large scale incremental learning. (CVPR 2019)
> >
> > [5] Incremental Classifier and Representation Learning. (CVPR 2017)
> >
> > [6] Maintaining Discrimination and Fairness in Class Incremental Learning. (CVPR 2020)
> >
> > [7] A theoretical study on solving continual learning. (NeurIPS2022)
> >
> > [8] Class-incremental learning: survey and performance evaluation on image classification. (TPAMI 2022)
> >
> > [9] Deep Class-Incremental Learning: A Survey
> >
> >
> > Q6: Please give the efficiency results.
> > -----------------
> > **A6:** Table 3 compares the efficiency across different methods using the TinyImageNet dataset, all under identical computing environments and settings (i.e. zero-base with 10 phases). Compared to all baselines, our methods require lower GPU memory usage, lower Multiply–Accumulate Operations (MACs) [1], and shorter training time per epoch.
> >
> > Table 3: Efficiency, One GPU NVIDIA RTX A6000, BS=256, M=10^6, G=10^9, T=10^12,
> > | Tiny, 0-10 | MACs | Parameters | MAX GPU-Memo | Time/Epoch| Epochs |
> > | -------- | -------- | -------- | -------- | -------- | -------- |
> > | YoooP+ | 571.26 G | 11.17 M | 9.86 GB | 21.07 s | 60 |
> > | YoooP| 571.26 G | 11.17 M | 9.53 GB | 15.60 s | 60 |
> > | PASS | 2.28 T | 11.18 M | >50 GB | 40.64 s | 101 |
> > | FeTril | 571.26 G | 11.17 M | 10.72 GB | 349.15 s | Task 0: 200, Others: 50 |
> > | SSRE | 633.31 G | 12.40 M | 13.23 GB | 50.17 s | 101 |
> > | LwF | 571.26 | 11.17 M | 9.53 GB | 16.90 s | Task 0: 200, Others: 250 |
> > | IL2A | 2.74 T | 11.17 M | >50 GB | 97.4 s | 101 |
> > |       |         |          |         |        |       |
> >
> > **References**
> >
> > [1] https://en.wikipedia.org/wiki/Multiply%E2%80%93accumulate_operation

---

> > > ### Author Response · Authors · 2023-11-21
> > > **Reminder to reviewer: Looking Forward to Your Further Comments and Feedback**
> > >
> > > Dear Reviewer eHC3
> > > We would like to thank you for taking the time to review our paper and for the insightful comments. We have addressed all the comments and suggestions you made. In particular, as suggested, we have **compared our method with some exemplar-based methods, explained where the forgetting occurs, and given the efficiency results in our response**. As we are approaching the midpoint of the discussion period, please kindly let us know if you have any additional concerns. We truly appreciate this opportunity to improve our work and shall be most grateful for any feedback you could give to us. If you do not have further questions, we are curious if you could consider raising our score. Thank you very much!

---

### Official Review · Reviewer_gcwm · 2023-10-30

**Soundness:** 2 fair
**Presentation:** 2 fair
**Contribution:** 2 fair
**Rating:** 5
**Confidence:** 4

**Summary:**

The authors proposed new non-exemplar prototype-based method for class-incremental learning (CIL) setting. Prototype are optimized with attentional mean-shit method, while training the new tasks, they are replayed for old classes. To mitigate forgetting the autors as well use distillation at the feature level (l2). During the training, model interpolation and partial freezing of the classifier is used. This method is called YoooP. Additionally, in this work authors proposed extension to it, when some improvement to prototype replay based on data augmentation is proposed. The authors compared their method on Cifar-100 and TinyImageNet in compariosn to few other exemplar-free CIL methods.

**Strengths:**

S1. CIL exemplar-free setting is a challenging setting, where most of the recent work put more interest to fight only with forgetting - starting with already good backbone (base-50% setting) and only fighting with forgetting (FeTrIL, SSRE). This work seems to be competitive in harder "base0" setting.

S2. Visualization part of the paper is good, well support written text. But still, not enough to provide all the info - seem W1.

S3. Interesting founding that PASS w/o Augh works better for some settings.

**Weaknesses:**

W1. The main weakness of this work is that after the reading the main paper the reader won't be able to reproduce and know exactly how the method works. The crucial part is in the appendix, i.e. Algorithm 1, where you see that we have S steps inside each epoch to calculate L_t, R iterations to calculate prototypes p_k, main loss - Arcface is mentioned in the implementation details and it hyper-parameter sigma etc.

W2. Some equations give even a wrong intuition how the loss is computed, e.g Eq.4. and the first component is over the whole dataset, while in practice is withing S minibatches.

W3. Some crucial hyper-parameters are not provided in main paper, or not provided at all - number of S steps, R iterations.

W4. Multiple things are combinend in the Yooop method  - three losses components, where feature distillation is quite strong regularization preventing forgetting but at the same time lowering plasticity. Additionally, mixing this with model interpolation with beta = 0.6 and freezing the classifier. The ablation is not clear about every single participant. Seems like the model interpolation is crucial here, and bug gain is for YoooP+.

W5. Why in the first figure we have missing units for the memory size? It's the first figure that supposed to give some motivation, but currently it raises more questions. Additionally, SSRE to my knowledge has the growing part and then compression. How it's possible that it's in line with FeTrIL, PASS, IL2A?

W6. Why the method is not evaluated with ImageNet-100? This dataset would shade more light how the method performs on the bigger images and how to compare with the other baselines.

W7. Logic in the reasoning. Page 7, before saying: "Therefore, we can draw a conclusion that the proposed methods can outperform the non-exemplar baselines." :

    While SSRE and FeTrIL have lower average forgetting than our methods, their prediction accuracy drops rapidly in the initial tasks, resulting in low accuracy in the final task,  as shown in Figure 4.  In reality,  the lower forgetting in SSRE and FeTrIL is attributed to the sharp drop in accuracy in the early tasks and the limited learning progress in the subsequent tasks

That is true about SSRE and FeTrIL, I agree. They mostly fight with forgetting. That's why they are good in base-half setting (see Tiny and FeTrIL in your appendix, Yooop is lower there). But how you can conclude this after this two sentences. Overall, in base0 setting your methods presents better accuracy. But to conclude this, the behavior of the SSRE FeTrIL dosen't matter here.

W8. Sec 4.3 "This is because YoooP+ can form a compact cluster for the samples in each class via prototype optimization and create high-quality synthetic data from the original distribution of cosine similarity to constrain **the boundary of old tasks.**"   - maybe the experiment to show that?

Overall, it is quite interesting work, but Yooop combines so many things, and it's not clear how each of it contributes. After reading the main paper you cannot re-produce it, you can have the idea how it works, appendix is necessary, but still not enough. The paper should be rewritten paying attention in all the hyper-params, mentioning arcface loss with sigma, and good ablation. Maybe some insight, comparison with PASS can be moved to the appendix (to have space for more crucial information/ablations).

**Questions:**

Q1. Out of the curiosity, why in the intro when you introduce class-incremental learning setting you ref. to three quite recent works of Zhu and Zhou (2021-2022)? This scenario of CL training is with us longer than that.

Q2. Why in the ablation of beta (Fig.8 (a)) we see different starting points for the first task? Beta is the interpolation that will take place after the first task.

Q3. Why starting points in Fig.4 for Yooop and Yooop+ are not the same? (easy to see for Tiny)

Q4. Why not providing source code in the supplementary? Without it I think the results will be hard to reproduce. I guess you use an existing framework for your work, looking at the results it's PyCIL? Would be good to give credits to authors and point your starting impoementations.

---

> ### Author Response · Authors · 2023-11-17
> **Response to Reviewer gcwm (Part 1/2)**
>
> Thank you for your constructive comments and suggestions! We have addressed all the comments and suggestions from the Reviewer and accordingly updated our manuscript highlighted in BLUE. We hope our responses below address your concerns. Please let us know if you have any additional concerns.
>
>
> Q1: Hard to reproduce with hyperparameters.
> -----------------
> **A1:** To reproduce our experimental results, we have uploaded the source code in the Supplementary Material. Additionally, we will make the source code publically available on GitHub upon publication. By the way, our source code is not based on an existing framework like PyCIL.
>
> Regarding parameters $S$ steps and $R$ iterations in Algorithm 1, they are determined by the ratio of Dataset Size to Mini-batch size. More specifically, $S$ is the number of total steps per epoch for each task. For the number of steps $R$ used to update the prototypes in each task, we simply set $R=S$ in our experiments. We have added the explanations of these two parameters in Appendix A in our revised version.
>
> For the Arcface loss hyperparameter $\delta$, we already conducted an ablation study in Appendix C.5 and consistently used $\delta=0.25$ across all experiments with various datasets. Per your suggestion, we have introduced the Arcface loss in the main paper.
>
> Q2: Eq.4 has the misunderstanding.
> -----------------
> **A2:** Per your suggestion, we added some explanations of Eq.4 in the updated version as follows.
>
> where $\ell(\cdot,\cdot)$ is a classification loss, and $i\in [n_t]:y_i=k$ is replaced by $i\in B$ when training by mini-batch.
>
> Q3: Some crucial hyper-parameters are not provided in main paper, or not provided at all - number of S steps, R iterations.
> -------------------
> **A3:** Per your suggestion, we have introduced the important hyper-parameters in the main paper. Regarding parameters $S$ steps and $R$ iterations, they are actually determined by the ratio of Dataset Size to Mini-batch size. Please refer to **A1** above.
>
>
> Q4: The ablation is not clear enough.
> -----------------
> **A4:** The classification loss $L_{t,C}$ and the knowledge distillation loss $L_{t,KD}$ in YoooP are commonly used in recent Non-exemplar CIL methods[1],[2],[3]. These two components are also adopted by the baselines. In addition, the partial freezing of the classifier is also used in existing work LwF[1]. So any improvement over the baselines is a result of the proposed $L_{t,P}$ loss.
>
> Accordingly, we do an ablation study on the newly proposed components, including model interpolation (MI), prototype optimization \(P\), and prototype augmentation (PA) in YoooP+. As illustrated in Figure 7 in Appendix C.1, we can see that YoooP(-P), which means without prototype optimization ($L_{t,P}$), will result in much lower accuracy, 51.50\%, than that of YoooP(-MI), 54.82\%. It thus suggests that prototype optimization is more important than MI. However, MI is also important since the accuracy of YoooP(-MI) will be lower compared to YoooP, 57.66\%.
>
> Additionally, we conducted an additional ablation study about MI on YoooP+. The evaluation results on CIFAR-100 with zeros-base 10 phases setting are reported in Table 1. Removing MI from YoooP+ reduces the accuracy by 3.91\% while removing PA from YoooP+ (i.e. YoooP) reduces the accuracy more, 4.27\%. Thus, prototype augmentation (PA) in YoooP+ is more important than MI.
>
> Table 1: Additional Ablation Study
> | CIFAR-100 | zero-10 Avg | zero-10 Last |
> | -------- | -------- | -------- |
> | YoooP | 57.66\% | 42.49\% |
> | YoooP (-MI) | 55.42\% | 37.2\% |
> | YoooP+ (-MI) | 58.02\% | 41.2\% |
> | YoooP+ (YoooP +PA) | 61.93\% | 47.17\% |
> | | | |
>
> **References**
>
> [1] Learning without Forgetting. (ECCV2016)
>
> [2] Prototype Augmentation and Self-Supervision for Incremental Learning. (CVPR2021)
>
> [3] Self-Sustaining Representation Expansion for Non-Exemplar Class-Incremental Learning. (CVPR2022)
>
> Q5: The first figure misses units for the memory size as well as SSRE to my knowledge has the growing part and then compression.
> -----------------
> **A5:** Thanks for your suggestion! We have updated the units for the memory size in Figure 1.
>
> As described in the caption of Figure 1 and detailed in Appendix D, the term 'memory size' denotes the memory footprint of our method and other approaches when computing prototypes for each task during training.
>
> For SSRE, our reported memory size does not refer to the GPU memory during model training. SSRE, IL2A, and PASS compute the mean value of all embeddings per class. Thus, they have the same memory footprint (same line in Figure 1) and incur an expensive memory footprint for storing their embeddings. In contrast, our proposed mini-batch attentional mean-shift method computes prototypes in a mini-batch manner. This significantly reduces the memory footprint required for prototype generation.

---

> > ### Author Response · Authors · 2023-11-21
> > **Reminder to reviewer: Looking Forward to Your Further Comments and Feedback**
> >
> > Dear Reviewer gcwm,
> >
> > We would like to thank you for taking the time to review our paper and for the insightful comments. We have addressed all the comments and suggestions you made. In particular, as suggested, we have **conducted additional experiments on ImageNet-100, implemented ablation studies of three components,  and explained the important parameters in our response**. As we are approaching the midpoint of the discussion period, please kindly let us know if you have any additional concerns. We truly appreciate this opportunity to improve our work and shall be most grateful for any feedback you could give to us. If you do not have further questions, we are curious if you could consider raising our score. Thank you very much!

---

> > > ### Author Response · Authors · 2023-11-22
> > > **We are looking forward to your response**
> > >
> > > Dear Reviewer gcwm,
> > >
> > > As we are approaching the end of the discussion period (in <20 hours), we would like to cordially inquire about the extent to which we have successfully addressed the concerns outlined in your review. In particular, as suggested, **we have conducted additional experiments on ImageNet-100, implemented ablation studies of three components and explained the important parameters in our response**.  Should any lingering points require further attention, please rest assured that we are enthusiastic about the opportunity to provide comprehensive responses to any subsequent queries or comments you may have.
> > >
> > > Your constructive input remains invaluable to us, and we appreciate your dedication to enhancing the quality of our manuscript. Thank you for your time and consideration.
> > >
> > > Best,
> > >
> > > Authors

---

> ### Author Response · Authors · 2023-11-17
> **Response to Reviewer gcwm (Part 2/2)**
>
> Q6: Evaluation on large ImageNet-100 dataset.
> -----------------
> **A6:** Per your suggestion, we have evaluated our methods on ImageNet-100. The results are reported in the following table, where "*" denotes non-exemplar methods, "-last" means the average accuracy of the last task. IL2A is not included in the comparison because we do not have enough GPU memory to run their model on ImageNet-100. Our methods (YoooP and YoooP+) still significantly outperform the baselines in terms of average accuracy on the new dataset.
>
> Table 2: Evaluation on ImageNet-100.
> |  ImageNet-100  | Half-10 Avg | Half-10 last |
> |  ----  | ----  | ---- |
> | * YoooP | 75.97\% | 68.96\% |
> | * YoooP+ | 77.64\% | 70.54\% |
> | * PASS | 64.3\% | 53.54\% |
> | * FeTril | 72.55\% | 64.06\% |
> | * SSRE | 60.23\% | 51.20\% |
> | * LWF | 47.35\% | 30.53\% |
> | iCaRL-CNN | 48.46\% | 36.29\% |
> | iCaRL-NME | 59.04\% | 48.69\% |
> | BiC | 50.69\% | 31.06\% |
> | WA | 61.04\% | 46.24\% |
> | | | |
>
> Q7: Logic in the reasoning.
> -----------------
> **A7:** We already updated the logic meaning on Page 7 as follows.
>
> According to Table 1 and Figure 4, both SSRE and FeTrIL have lower forgetting while the prediction accuracy drops rapidly in the initial tasks. A lower forgetting in this case (with lower accuracy) indicates that the model is not improving or learning so such performance is not preferred. In contrast, the proposed YoooP and YoooP+ reach higher average incremental accuracy while achieving slightly higher or comparable forgetting compared to other non-exemplar baselines. This indicates a better stability-plasticity trade-off, implying that the model learns quickly on new tasks while incurring a minimal cost in forgetting. Therefore, it can be concluded that the proposed methods outperform the non-exemplar baselines.
>
> Q8: Synthetic data constrain the boundary of old tasks.
> -----------------
> **A8:** Our experiments in **Figure 6** verify that our method can achieve better classification boundaries for old tasks. As shown in Figure 6, the light gray points represent the distribution of data from old tasks. A comparison between Figure 6(a) and Figure 6(b) reveals that the data from previous tasks are still clustered into several groups using our method, exhibiting a clear boundary with the clusters of current task data. This pattern is also evident in Figure 6\(c\). The high-quality synthetic data effectively constrains the boundary of old tasks when training the current tasks, enabling the current tasks to establish a clear boundary with previous tasks. In contrast, the baselines like PASS (illustrated in Figure 6(d), (e), (f)), without our proposed prototype augmentation in YoooP+, will make the boundary between the current task data and previous task data gradually become less clear and their classes may even overlap with each other.
>
>
> Q9: References in the Introducion.
> -----------------
> **A9:** We have cited some earlier works on class-incremental learning in the Introduction in our revised version.
>
> Q10: Different starting points in Fig.8(a).
> -----------------
> **A10:** According to Equation 12, we update the parameters $\theta$ based on $\beta$. The different $\beta$ in our ablation study result in different starting points, $\theta_1$, for each case.
>
> Q11: Different start points for YoooP and YoooP+ in Fig.4.
> -----------------
> **A11:** It is caused by the random error (random seeds) in the experiments. Below, we present the exact accuracy of the first task in Figure 4. We can see that their accuracy is very close to each other.
>
> Table 3: First Task Accuracy
> | CIFAR-100 | zero-5 | zero-10 |
> | -------- | -------- | -------- |
> | YoooP | 87.67\% | 92.30\% |
> | YoooP+| 88.43\% | 91.68\% |
> | **TinyImageNet** | **zero-5** | **zero-10** |
> | YoooP | 79.48\% | 84.77\% |
> | YoooP+| 80.53\% | 86.05\% |
> |      |      |      |

---

### Official Review · Reviewer_ZsAp · 2023-10-31

**Soundness:** 2 fair
**Presentation:** 2 fair
**Contribution:** 3 good
**Rating:** 6
**Confidence:** 4

**Summary:**

(Motivation)
The prototype-based (class-mean) method, which is one of the exemplar-free Class Incremental Learning (CIL) methods, stores class-representative prototypes (=class mean prototype). However, the class mean prototype does not represent the centroids of each class. Utilizing inappropriate prototypes leads to confusion between old classes learned in different stages. Moreover, prototype augmentation, which adds Gaussian noise to the prototype, leads to more serious catastrophic forgetting of previously observed classes.

(Method)
1. They propose the prototype-based CIL method, called YoooP, that optimizes the prototype, considering the weighted average distance of all samples in the class. The paper claims this class-wise prototype is more representative than utilizing the simple class mean one.

2. They propose a prototype augmentation, called YoooP+, that synthesizes the prototypes from the angular distribution between each class’s real data and prototype stored in memory. Synthetic prototypes preserve the distribution of the real data rather than simply adding the Gaussian noise to the prototype (augmentation technique of PASS).

**Strengths:**

This paper proposes a novel approach to leave the class representative prototype considering the similarity with all real samples. The proposed method aims to be an efficient technique that can be easily applied to the existing prototype-based method.

This paper also proposes a data augmentation performed with the help of a rotation matrix. This strategy approximates real distribution better than conventional prototype augmentation techniques.

In base-0 experiments on CIFAR-100 and Tiny-ImageNet, the proposed methods achieve the best performance.

**Weaknesses:**

-	The scale of dataset (CIFAR-100, TinyImageNet) is small. Whether the proposed method works with bigger datasets is an important issue that needs to be addressed.
-	The paper lacks the result with 20 phases which is a quite common configuration.
-	The effect of an optimized prototype based on the attentional mean-shift method seems limited in many situations.

**Questions:**

1) For verifying the strength of proposed augmentation, why don’t you apply the new strategy to PASS instead of adding Gaussian noise to prototype? Would you show the performance of it?

2) Could you show the classification confusion matrix result of your approach?

3) In section 3, when define the probabilities over all classes C_(1:t ), isn’t softmax(wG(F(x;θ))) more accurate notation than softmax(wF(x;θ))?

4) The function c(,) in equation 2, equation 4 seems to be different. Is the c(,) in equation 2 cosine similarity function and one in equation 4 classifier (=G(F(p_k;θ)))?

5) How can we define the initial class-wise prototype p_k? Is it the same with the prototype used in PASS? (k-th class mean prototype)?

---

> ### Author Response · Authors · 2023-11-17
> **Response to Reviewer ZsAp (Part 1/2)**
>
> Thank you for your constructive comments and suggestions! We have addressed all the comments and suggestions from the Reviewer and accordingly updated our manuscript highlighted in BLUE. We hope our responses below address your concerns. Please let us know if you have any additional concerns.
>
>
> Q1: Whether the method works with bigger datasets
> ------------------
> **A1:** Thanks for your suggestion! We have evaluated the proposed two approaches on ImageNet-100. The results are reported in the following table, where "*" indicates non-exemplar methods, "-last" means the average accuracy of the last task. IL2A is not included in the comparison because we do not have enough GPU memory to train their model on ImageNet-100. Our methods (YoooP and YoooP+) still significantly outperform the baselines in terms of average accuracy on the large ImageNet-100 dataset.
>
> Table 1: Evaluation on ImageNet-100.
> |  ImageNet-100  | Half-10 Avg | Half-10 last |
> |  ----  | ----  | ---- |
> | * YoooP | 75.97\% | 68.96\% |
> | * YoooP+ | 77.64\% | 70.54\% |
> | * PASS | 64.3\% | 53.54\% |
> | * FeTril | 72.55\% | 64.06\% |
> | * SSRE | 60.23\% | 51.20\% |
> | * LWF | 47.35\% | 30.53\% |
> | iCaRL-CNN | 48.46\% | 36.29\% |
> | iCaRL-NME | 59.04\% | 48.69\% |
> | BiC | 50.69\% | 31.06\% |
> | WA | 61.04\% | 46.24\% |
> |  |  |   |
>
>
> Q2: About the result with 20 phases
> ------------------
> **A2:** We have evaluated the proposed methods on TinyImageNet under zero-base 20 phases and half-base 20 phases settings. The results are reported in the following table, where "*" indicates non-exemplar methods, "-last" means the average accuracy of the last task. We can observe that our methods (YoooP and YoooP+) still significantly outperform the baselines in terms of average accuracy on the 20 phases settings.
>
> Table 2: Evaluation on TinyImageNet.
> |  TinyImageNet  | Zero-20 Avg | Zero-20 last | Half-20 Avg | Half-20 last |
> |  ----  | ----  | ---- | ----  | ---- |
> | * YoooP | 46.00\% | 30.97\% | 48.19\% | 33.31\% |
> | * YoooP+ | **48.36%** | **31.26\%** | **51.27\%** | **35.28\%** |
> | * PASS | 33.76\% | 19.36\% | 33.24\% | 25.21\% |
> | * IL2A | 25.39\% | 17.73\% | 39.9\% | 29.43\% |
> | * FeTril | 31.84\% | 18.38\% | 48.96\% | 41.44\% |
> | * SSRE | 25.78\% | 15.25\% | 46.2\% | 38.84\% |
> | * LWF | 22.93\% | 9.57\% | 13.39\% | 5.62\%
> | iCaRL-CNN | 30.33\% | 11.31\% | 28.9\% | 20.54\% |
> | iCaRL-NME | 36.07\% | 18.35\% | 33.68\% | 27.3\% |
> | BiC | 34.29\% | 13.14\% | 40.81\% | 19.91\% |
> | WA | 35.87\% | 13.17\% | 27.93\% | 11.91\% |
> |    |  |    |  |  |
>
> Q3: Optimized prototype seems limited in situations.
> -----------------
> **A3:** Our prototype optimization method is versatile and can be applied not only to prototype-based methods but also to various scenarios in class incremental learning, as demonstrated in [1][2][3][4].
>
> **References**
>
> [1] Incremental Classifier and Representation Learning. (CVPR 2017)
>
> [2] Large Scale Incremental Learning. (CVPR 2019)
>
> [3] BEEF: Bi-Compatible Class-Incremental Learning via Energy-Based Expansion and Fusion. (ICLR 2023)
>
> [4] Feature Boosting and Compression for Class-incremental Learning. (ECCV 2022)
>
>
> Q4: Apply the new augmentation strategy to PASS.
> -----------------
> **A4:** We have applied our new prototype augmentation (PA) strategy to the PASS, and then evaluated its performance on CIFAR-100 under the zero-base 10 phases setting. The results are reported in the following table. We can observe that our prototype augmentation method can significantly improve the performance of PASS, even without prototype optimization.
>
> Table 3: Comparison of adding Gaussian noise and our strategy on PASS.
> | CIFAR-100 | zero-10 Avg | zero-10 Last |
> | -------- | -------- | -------- |
> | PASS (Adding noise)  | 51.94\% | 35.81\% |
> | PASS with PA | 55.98\% | 41.03\% |
>
>
> Q5: Show the classification confusion matrix.
> -----------------
> **A5:** Thanks for your suggestion! We randomly choose 10 classes to show a part of the classification confusion matrix of YoooP+, and the entire confusion matrix of 100 classes is presented in Appendix E of the revised version.
>
> Table 4: Confusion Matrix (randomly sampled 10x10 submatrix) for CIFAR-100 with zero-base 10 phases setting
> | Confusion Matrix | Road | Lobster | Cattle | Couch | Shrew | Aquarium fish | Castle | Palm tree | Snail | Bowl |
> | -------- | -------- | -------- | -------- | -------- | -------- | -------- | -------- | -------- | -------- | -------- |
> | **Road** | 79 | 0 | 0 | 0 | 0 | 0 | 0 | 0 | 0 | 0 |
> | **Lobster** | 0 | 76 | 0 | 0 | 0 | 0 | 0 | 0 | 0 | 0 |
> | **Cattle** | 1 | 0 | 58 | 0 | 0 | 0 | 0 | 0 | 0 | 0 |
> | **Couch** | 0 | 0 | 0 | 48 | 0 | 0 | 0 | 0 | 0 | 0 |
> | **Shrew** | 0 | 0 | 6 | 0 | 54 | 0 | 0 | 0 | 0 | 0 |
> | **Aquarium fish** | 0 | 0 | 0 | 0 | 0 | 54 | 0 | 0 | 0 | 0 |
> | **Castle** | 0 | 0 | 0 | 0 | 0 | 0 | 24 | 0 | 0 | 0 |
> | **Palm tree** | 0 | 1 | 0 | 1 | 0 | 0 | 0 | 61 | 0 | 0 |
> | **Snail** | 0 | 1 | 0 | 0 | 0 | 0 | 0 | 0 | 41 | 0 |
> | **Bowl** | 0 | 0 | 0 | 0 | 0 | 0 | 1 | 0 | 0 | 44 |

---

> ### Author Response · Authors · 2023-11-17
> **Response to Reviewer ZsAp (Part 2/2)**
>
> Q6: More accurate notation.
> -----------------
> **A6:** Thanks for your suggestion! We have already updated the notation with $softmax(G(F(x; \theta); w))$ in Section 3 highlighted in blue.
>
>
> Q7: The function c(,) in Eq.2, Eq.4.
> -----------------
> **A7:** Thanks for your correction! After fixing Eq.4 in our revised version, $c(\cdot,\cdot)$ in both Eq. 2 and Eq.4 denotes the cosine similarity function. In Eq. 4, it is used to compute the logits for all the classes $j\in C_{1:t}$ as the cosine similarities between each vector in the weight matrix of the final-layer classifier and (1) a sample's representation $z_i$ (the first term in Eq. 4); or (2) each class's prototype $p_k$ (the second term in Eq. 4).
>
> $L_{t,C}(\theta,w)\triangleq\frac{1}{|C_t|}\frac{1}{n_t}\sum_{k\in C_t}\sum_{i\in [n_t]:y_i=k}\ell([c(z_i, w_j)]\_{j\in C_{1:t}, k)} + \frac{1}{|C_{1:{t-1}}|}\sum_{k\in C_{1:{t-1}}}\ell([c(p_k, w_j)]\_{j\in C_{1:t}}, k)$
>
>
> Q8: The initial class-wise prototype $p_k$.
> -----------------
> **A8:** We randomly choose one sample from each class-$k$ and use its embedding as the initial prototype $p_k$. Our prototype update method is an extension of the classical mean-shift algorithm [1] that replaces the pre-defined kernel similarity with a learnable attention score. As stated in [1], random initialization of $p_k$ can guarantee the independence of the solution to the initialization problem.
>
> **References**
>
> [1] Mean shift, mode seeking, and clustering, TPAMI 1995.

---

> > ### Author Response · Authors · 2023-11-21
> > **Reminder to reviewer: Looking Forward to Your Further Comments and Feedback**
> >
> > Dear Reviewer ZsAp
> >
> > We would like to thank you for taking the time to review our paper and for the insightful comments. We have addressed all the comments and suggestions you made. In particular, as suggested, we have **conducted new experiments on another large ImageNet-100 dataset and evaluated the proposed methods on TinyImageNet under zero-base 20 phases and half-base 20 phases settings in our response**. As we are approaching the midpoint of the discussion period, please kindly let us know if you have any additional concerns. We truly appreciate this opportunity to improve our work and shall be most grateful for any feedback you could give to us. If you do not have further questions, we are curious if you could consider raising our score. Thank you very much!

---

> > > ### Comment · Reviewer_ZsAp · 2023-11-22
> > >
> > > I read the rebuttal carefully and thank the authors for carrying out the experiments we asked for.
> > > I still have concerns with the effectiveness of the proposed method on a small dataset and cannot entirely agree the proposed optimized prototype represents each class's value.
> > > So, I will keep my decision (marginally above the acceptance).

---

> > > > ### Author Response · Authors · 2023-11-23
> > > > **Response to the follow up questions by Reviewer ZsAp**
> > > >
> > > > Thank you for your response! We are glad to hear that some of your concerns have been successfully addressed! We hope the following explanation can address your futher concerns! Please let us know if you have any remaining concerns and we are willing to have more discussion.
> > > >
> > > > Q1: The effectiveness on a small dataset
> > > > -----------------
> > > > **A1:** As illustrated in Figure 4 and Table 1, our proposed methods demonstrate excellent performance on small datasets such as CIFAR-100 and TinyImageNet. Specifically, under the zero-base 5 phases setting, YoooP+ outperforms PASS by **6.93\%** in average incremental accuracy on CIFAR-100, and under the zero-base 10 phases setting, it surpasses PASS by **9.99\%**. On TinyImageNet, YoooP+ achieves an average incremental accuracy increase of **14.08\%** in the zero-base 5 phases setting and **17.85\%** in the zero-base 10 phases setting compared to PASS. Consequently, we can conclude that our method, YoooP+, is effective for small datasets.
> > > >
> > > > Q2: Cannot entirely agree the optimized prototype represents each class's value
> > > > -----------------
> > > > **A2:** In order to address that futher concerns, we also conducted an extra ablation experiment. We use the class-mean as our prototype without prototype optimization to compare wth the performance with YoooP, the results are shown as following table. The performance of YoooP surpass the performance of YoooP using class-mean without prototype optimization. Thus, the results can indirectly proof that the optimized prototype represents each class's value better.
> > > >
> > > > Table 1: Evaluation on CIFAR-100.
> > > > |  CIFAR-100  | zero-10 Avg | zero-10 last |
> > > > |  ----  | ----  | ---- |
> > > > | YoooP | 57.66\% | 42.49\% |
> > > > | YoooP (class-mean w/o optimization) | 48.30\% | 30.66\% |
> > > > | | | |

---

### Meta-Review · Area_Chair_8Bao · 2023-12-19

**Metareview:**

This paper builds on prototype-based methods for continual learning, specifically proposing to optimize them with an attentional mean-shift method and utilizing them for replay during task training. Further, an augmentation method is proposed using sampling from an angular distribution. Results are shown on standard datasets such as CIFAR-100 and TinyImageNet.

  While the reviewers appreciated the reasonable performance and design of the method, along with clear writing, the paper received mixed scores with significant concerns. This included the scale of the experimentation (datasets, sequence size, etc.), reproducibility of the paper due to missing details such as hyper-parameters, novelty of the method given the number of prototype-based methods (whereas this paper adds the mean-shift idea and augmentation), lack of thorough comparison to relevant prior strong methods, and overall lack of clarity. While the authors provided a rebuttal, including with some additional results, the reviewers that were most negative were still not convinced. Notably, the question about whether contrastive learning could equally do what the proposed algorithm effectively does and the additional strong baselines were not strongly addressed; for example, the authors added additional methods that were similar in characteristics to the ones suggested by the reviewer but not as strong.

  After considering the paper, the reviews (which are mixed at 3,5,6,6), the rebuttals, and subsequent discussion, this paper is currently not sufficiently strong enough in its contribution and execution/thoroughness to warrant acceptance. I recommend that the authors strengthen these aspects for resubmission.

**Justification For Why Not Higher Score:**

Overall, the paper has a number of deficiencies across the spectrum: clarity/reproducibility, technical contribution over prior methods, and experimental comparisons.

**Justification For Why Not Lower Score:**

N/A

---

### Decision · Program_Chairs · 2024-01-16

Reject